# Unlocking neural population non-stationarity using a hierarchical dynamics model

**Mijung Park**[1]**, Gergo Bohner**[1]**, Jakob H. Macke**[2]
1 Gatsby Computational Neuroscience Unit, University College London
[2] Research Center caesar, an associate of the Max Planck Society, Bonn
Max Planck Institute for Biological Cybernetics,
Bernstein Center for Computational Neuroscience Tübingen
{mijung, gbohner}@gatsby.ucl.ac.uk, jakob.macke@caesar.de

## Abstract

Neural population activity often exhibits rich variability. This variability can arise from single-neuron stochasticity, neural dynamics on short time-scales, as well as from modulations of neural firing properties on long time-scales, often referred to as neural non-stationarity. To better understand the nature of co-variability in neural circuits and their impact on cortical information processing, we introduce a hierarchical dynamics model that is able to capture both slow inter-trial modulations in firing rates as well as neural population dynamics. We derive a Bayesian Laplace propagation algorithm for joint inference of parameters and population states. On neural population recordings from primary visual cortex, we demonstrate that our model provides a better account of the structure of neural firing than stationary dynamics models.

## 1 Introduction

Neural spiking activity recorded from populations of cortical neurons can exhibit substantial variability in response to repeated presentations of a sensory stimulus [1]. This variability is thought to arise both from dynamics generated endogenously within the circuit [2] as well as from variations in internal and behavioural states [3, 4, 5, 6, 7]. An understanding of how the interplay between sensory inputs and endogenous dynamics shapes neural activity patterns is essential for our understanding of how information is processed by neuronal populations. Multiple statistical [8, 9, 10, 11, 12, 13] and mechanistic [14] models for characterising neuronal population dynamics have been developed.

In addition to these dynamics which take place on fast time-scales (milliseconds up to few seconds), there are also processes modulating neural firing activity which take place on much slower time-scales (seconds to hours). Slow drifts in rates across an experiment can be caused by fluctuations in arousal, anaesthesia level or other physiological properties of the experimental preparation [15, 16, 17]. Furthermore, processes such as learning and short-term plasticity can lead to slow changes in neural firing properties [18]. The statistical structure of these slow fluctuations has been modelled using state-space models and related techniques [19, 20, 21, 22, 23]. Recent experimental findings have shown that slow, multiplicative fluctuations in neural excitability are a dominant source of neural covariability in extracellular multi-cell recordings from cortical circuits [5, 17, 24].

To accurately capture the the structure of neural dynamics and to disentangle the contributions of slow and fast modulatory processes to neural variability and co-variability, it is therefore important to develop models that can capture neural dynamics both on fast (i.e., within experimental trials) and slow (i.e., across trials) time-scales. Few such models exist: Czanner et al. [25] presented a statistical model of single-neuron firing in which within-trial dynamics are modelled by (generalised) linear coupling from the recent spiking history of each neuron onto its instantaneous firing rate, and across-trial dynamics were modelled by defining a random walk model over parameters. More recently,

Mangion et al [26] presented a latent linear dynamical system model with Poisson observations (PLDS, [8, 11, 13]) with a one-dimensional latent space, and used a heuristic filtering approach for tracking parameters, again based on a random-walk model. Rabinowitz et al [27] presented a technique for identifying slow modulatory inputs from the recordings of single neurons using a Gaussian Process model and an efficient inference technique using evidence optimisation.

Here, we present a hierarchical model that consists of a latent dynamical system with Poisson observations (PLDS) to model neural population dynamics, combined with a Gaussian process (GP) [28] to model modulations in firing rates or model-parameters across experimental trials. The use of an exponential nonlinearity implies that latent modulations have a multiplicative effect on neural firing rates. Compared to previous models using random walks over parameters, using a GP is a more flexible and powerful way of modelling the statistical structure of non-stationarity, and makes it possible to use hyper-parameters that model the variability and smoothness of parameter-changes across time.

In this paper, we focus on a concrete variant of this general model: We introduce a new set of variables which control neural firing rate on each trial to capture non-stationarity in firing rates. We derive a Bayesian Laplace propagation method for inferring the posterior distributions over the latent variables and the parameters from population recordings of spiking activity. Our approach generalises the 1-dimensional latent states in [26] to models with multi-dimensional states, as well as to a Bayesian treatment of non-stationarity based on Gaussian Process priors. The paper is organised as follows: In Sec. 2, we introduce our framework for constructing non-stationary neural population models, as well as the concrete model we will use for analyses. In Sec. 3, we derive the Bayesian Laplace propagation algorithm. In Sec. 4, we show applications to simulated data and neural population recordings from visual cortex.

## 2 Hierarchical non-stationary models of neural population dynamics

We start by introducing a hierarchical model for capturing short time-scale population dynamics as well as long time-scale non-stationarities in firing rates. Although we use the term "non-stationary" to mean that the system is best described by parameters that change over time (which is how the term is often used in the context of neural data analysis), we note that the distribution over parameters can be described by a stochastic process which might be strictly stationary in the statistical sense[1].

**Modelling framework** We assume that the neural population activity of $p$ neurons $\mathbf{y}_t \in \mathbb{R}^p$ depends on a $k$-dimensional latent state $\mathbf{x}_t \in \mathbb{R}^k$ and a modulatory factor $\mathbf{h}^{(i)} \in \mathbb{R}^k$ which is different for each trial $i = \{1, \ldots, r\}$. The latent state $\mathbf{x}$ models short-term co-variability of spiking activity and the modulatory factor $\mathbf{h}$ models slowly varying mean firing rates across experimental trials.

We model neural spiking activity as conditionally Poisson given the latent state $\mathbf{x}_t$ and a modulator $\mathbf{h}^{(i)}$, with a log firing rate which is linear in parameters and latent factors,

$$\mathbf{y}_t | \mathbf{x}_t, C, \mathbf{h}^{(i)}, \mathbf{d} \sim \text{Poiss}(\mathbf{y}_t | \exp(C(\mathbf{x}_t + \mathbf{h}^{(i)}) + \mathbf{d})),$$

where the loading matrix $C \in \mathbb{R}^{p \times k}$ specifies how each neuron is related to the latent state and the modulator, $\mathbf{d} \in \mathbb{R}^p$ is an offset term that controls the mean firing rate of each cell, and $\text{Poiss}(\mathbf{y}_t | \mathbf{w})$ means that the $i$th entry of $\mathbf{y}_t$ is drawn independently from Poisson distribution with mean $\mathbf{w}_i$ (the $i$th entry of $\mathbf{w}$). Because of the use of an exponential firing-rate nonlinearity, latent factors have a multiplicative effect on neural firing rates, as has been observed experimentally [17, 5].

Following [11, 13, 26], we assume that the latent dynamics evolve according to a first-order autoregressive process with Gaussian innovations,

$$\mathbf{x}_t | \mathbf{x}_{t-1}, A, B, Q \sim \mathcal{N}(\mathbf{x}_t | A\mathbf{x}_{t-1} + B\mathbf{u}_t, Q).$$

Here, we allow for sensory stimuli (or experimental covariates), $\mathbf{u}_t \in \mathbb{R}^d$ to influence the latent states linearly. The dynamics matrix $A \in \mathbb{R}^{k \times k}$ determines the state evolution, $B \in \mathbb{R}^{k \times d}$ models the dependence of latent states on external inputs, and $Q \in \mathbb{R}^{k \times k}$ is the covariance of the innovation noise. We set $Q$ to be the identity matrix, $Q = \mathbf{I}_k$ as in [29], and we assume $\mathbf{x}_0^{(i)} \sim \mathcal{N}(0, \mathbf{I}_k)$.

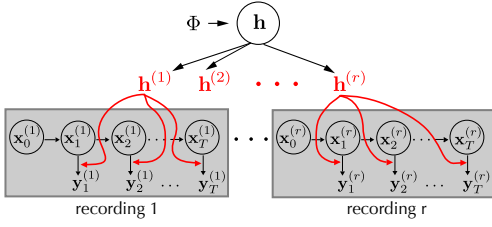

Figure 1: Schematic of hierarchical non-stationary Poisson observation Latent Dynamical System (N-PLDS) for capturing non-stationarity in mean firing rates. The parameter $\mathbf{h}$ slowly varies across trials and leads to fluctuations in mean firing rates.

The parameters in this model are $\boldsymbol{\theta} = \{A, B, C, \mathbf{d}, \mathbf{h}^{(1:r)}\}$. We refer to this general model as *non-stationary PLDS* (N-PLDS). Different variants of N-PLDS can be constructed by placing priors on individual parameters which allow them to vary across trials (in which case they would then depend on the trial index $i$) or by omitting different components of the model[2].

For the modulator $\mathbf{h}$, we assume that it varies across trials according to a GP with mean $\mathbf{m_h}$ and (modified) squared exponential kernel, $\mathbf{h}^{(i)} \sim \mathcal{GP}(\mathbf{m_h}, K(i,j))$, where the $(i,j)$th block of $K$ (size $k \times k$) is given by $K(i,j) = (\sigma^2 + \epsilon\delta_{i,j})\exp\left(-\frac{1}{2\tau^2}(i-j)^2\right)I_k$. Here, we assume the independent noise-variance on the diagonal ($\epsilon$) to be constant and small as in [30]. When $\sigma^2 = \epsilon = 0$, the modulator vanishes, which corresponds to the conventional PLDS model with fixed parameters [11, 13]. When $\sigma^2 > 0$, the mean firing rates vary across trials, and the parameter $\tau$ determines the time-scale (in units of 'trials') of these fluctuations. We impose ridge priors on the model parameters (see Appendix for details), so that the total set of hyperparameters of the model is $\Phi = \{\mathbf{m_h}, \sigma^2, \tau^2, \boldsymbol{\phi}\}$, where $\boldsymbol{\phi}$ is the set of ridge parameters.

## 3 Bayesian Laplace propagation

Our goal is to infer parameters and latent variables in the model. The exact posterior distribution is analytically intractable due to the use of a Poisson likelihood, and we therefore assume the joint posterior over the latent variables and parameters to be factorising,

$$p(\boldsymbol{\theta}, \mathbf{x}_{1:T}^{(1:r)}|\mathbf{y}_{1:T}^{(1:r)}, \Phi) \propto p(\mathbf{y}_{1:T}^{(1:r)}|\mathbf{x}_{1:T}^{(1:r)}, \boldsymbol{\theta})p(\mathbf{x}_{1:T}^{(1:r)}|\boldsymbol{\theta}, \Phi)p(\boldsymbol{\theta}|\Phi) \approx q(\boldsymbol{\theta}, \mathbf{x}_{1:T}^{(1:r)}) = q_{\boldsymbol{\theta}}(\boldsymbol{\theta})\prod_{i=1}^{r}q_{\mathbf{x}}(\mathbf{x}_{0:T}^{(i)}).$$

This factorisation simplifies computing the integrals involved in calculating a bound on the marginal likelihood of the observations,

$$
\begin{aligned}
\log p(\mathbf{y}_{1:T}^{(1:r)}|\Phi) &= \log \int d\boldsymbol{\theta} \; d\mathbf{x}_{1:T}^{(1:r)} \; p(\boldsymbol{\theta}, \mathbf{x}_{1:T}^{(1:r)}, \mathbf{y}_{1:T}^{(1:r)}|\Phi), \\
&\geq \int d\boldsymbol{\theta} \; d\mathbf{x}_{1:T}^{(1:r)} \; q(\boldsymbol{\theta}, \mathbf{x}_{1:T}^{(1:r)}) \; \log \frac{p(\boldsymbol{\theta}, \mathbf{x}_{1:T}^{(1:r)}, \mathbf{y}_{1:T}^{(1:r)}|\Phi)}{q(\boldsymbol{\theta}, \mathbf{x}_{1:T}^{(1:r)})}.
\end{aligned}
\tag{1}
$$

Similar to variational Bayesian expectation maximization (VBEM) algorithm [29], our inference procedure consists of the following three steps: (1) we compute the approximate posterior over latent variables $q_{\mathbf{x}}(\mathbf{x}_{0:T}^{(1:r)})$ by integrating out the parameters

$$q_{\mathbf{x}}(\mathbf{x}_{0:T}^{(1:r)}) \propto \exp\left[\int d\boldsymbol{\theta} q_{\boldsymbol{\theta}}(\boldsymbol{\theta}) \log p(\mathbf{x}_{1:T}^{(1:r)}, \mathbf{y}_{1:T}^{(1:r)}|\boldsymbol{\theta})\right], \tag{2}$$

which is performed by forward-backward message passing relying on the order-1 dependency in latent states. Then, (2) we compute the approximate posterior over parameters $q_{\boldsymbol{\theta}}(\boldsymbol{\theta})$ by integrating out the latent variables,

$$q_{\boldsymbol{\theta}}(\boldsymbol{\theta}) \propto p(\boldsymbol{\theta}) \exp\left[\int d\mathbf{x}_{0:T}^{(1:r)} q_{\mathbf{x}}(\mathbf{x}_{0:T}^{(1:r)}) \log p(\mathbf{x}_{0:T}^{(1:r)}, \mathbf{y}_{1:T}^{(1:r)}|\boldsymbol{\theta})\right], \tag{3}$$

and (3) we update the hyperparameters by computing the gradients of the bound on the eq. 1 after integrating out both latent variables and parameters. We iterate the three steps until convergence.

Unfortunately, the integrals in both eq. 2 and eq. 3 are not analytically tractable, even with the Gaussian distributions for $q_{\mathbf{x}}(\mathbf{x}_{0:T}^{(1:r)})$ and $q_{\boldsymbol{\theta}}(\boldsymbol{\theta})$. For tractability and fast computation of messages in

the forward-backward algorithm for eq. 2, we utilise the so-called *Laplace propagation* or Laplace expectation propagation (Laplace-EP) [31, 32, 33], which makes a Gaussian approximation to each message based on Laplace approximation, then propagates the messages forward and backward. While Laplace propagation in the prior work is commonly coupled with point estimates of parameters, we consider the posterior distribution over parameters. For this reason, we refer to our inference method as *Bayesian Laplace propagation*. The use of approximate message passing in the Laplace propagation implies that there is no longer a guarantee that the lower bound will increase monotonically in each iteration, which is the main difference between our method and the VBEM algorithm. We therefore monitored the convergence of our algorithm by computing one-step ahead prediction scores [13]. The algorithm proceeds by iterating the following three steps:

**(1) Approximating the posterior over latent states:** Using the first-order dependency in latent states, we derive a sequential forward/backward algorithm to obtain $q_{\mathbf{x}}(\mathbf{x}_{0:T}^{(1:r)})$, generalising the approach of [26] to multi-dimensional latent states. Since this step decouples across trials, it is easy to parallelize, and we omit the trial-indices for clarity. We note that computation of the approximate posterior in this step is not more expensive than Bayesian inference of the latent state in a 'fixed parameter' PLDS. The forward message $\alpha(\mathbf{x}_t)$ at time $t$ is given by

$$\alpha(\mathbf{x}_t) \propto \int d\mathbf{x}_{t-1} \alpha(\mathbf{x}_{t-1}) \exp\left[\langle\log(p(\mathbf{x}_t|\mathbf{x}_{t-1})p(\mathbf{y}_t|\mathbf{x}_t))\rangle_{q_{\boldsymbol{\theta}}(\boldsymbol{\theta})}\right]. \tag{4}$$

Assuming that the forward message at time $t-1$ denoted by $\alpha(\mathbf{x}_{t-1})$ is Gaussian, the Poisson likelihood term will render the forward message at time $t$ non-Gaussian, but we will approximate $\alpha(\mathbf{x}_t)$ as a Gaussian using the first and second derivatives of the right-hand side of eq. 4 with respect to $\mathbf{x}_t$.

Similarly, the backward message at time $t-1$ is given by

$$\beta(\mathbf{x}_{t-1}) \propto \int d\mathbf{x}_t \beta(\mathbf{x}_t) \exp\left(\langle\log(p(\mathbf{x}_t|\mathbf{x}_{t-1})p(\mathbf{y}_t|\mathbf{x}_t))\rangle_{q_{\boldsymbol{\theta}}(\boldsymbol{\theta})}\right), \tag{5}$$

which we also approximate to a Gaussian for tractability in computing backward messages.

Using the forward/backward messages, we compute the posterior marginal distribution over latent variables (See Appendix). We need to compute the cross-covariance between neighbouring latent variables to obtain the sufficient statistics of latent variables (which we will need for updating the posterior over parameters). The pairwise marginals of latent variables are given by

$$p(\mathbf{x}_t, \mathbf{x}_{t+1}|\mathbf{y}_{1:T}) \quad \propto \quad \beta(\mathbf{x}_{t+1}) \exp\left(\langle\log(p(\mathbf{y}_{t+1}|\mathbf{x}_{t+1})p(\mathbf{x}_{t+1}|\mathbf{x}_t))\rangle_{q_{\boldsymbol{\theta}}(\boldsymbol{\theta})}\right) \alpha(\mathbf{x}_t), \tag{6}$$

which we approximate as a joint Gaussian distribution by using the first/second derivatives of eq. 6 and extracting the cross-covariance term from the joint covariance matrix.

**(2) Approximating the posterior over parameters:** After inferring the posterior over latent states, we update the posterior distribution over the parameters. The posterior over parameters factorizes as

$$q_{\boldsymbol{\theta}}(\boldsymbol{\theta}) \quad = \quad q_{\mathbf{a},\mathbf{b}}(\mathbf{a}, \mathbf{b}) \, q_{\mathbf{c},\mathbf{d},\mathbf{h}}(\mathbf{c}, \mathbf{d}, \mathbf{h}^{(1:r)}), \tag{7}$$

where used the vectorized notations $\mathbf{b} = \text{vec}(B^\top)$ and $\mathbf{c} = \text{vec}(C^\top)$. We set $\mathbf{c}, \mathbf{d}$ to the maximum likelihood estimates $\hat{\mathbf{c}}, \hat{\mathbf{d}}$ for simplicity in inference. The computational cost of this algorithm is dominated by the cost of calculating the posterior distribution over $\mathbf{h}^{(1:r)}$, which involves manipulation of a $rk$-dimensional Gaussian. While this was still tractable without further approximations for the data-set sizes used in our analyses below (hundreds of trials), a variety of approximate methods for GP-inference exist which could be used to improve efficiency of this computation. In particular, we will typically be dealing with systems in which $\tau \gg 1$, which means that the kernel-matrix is smooth and could be approximated using low-rank representations [28].

**(3) Estimating hyperparameters:** Finally, after obtaining the the approximate posterior $q(\boldsymbol{\theta}, \mathbf{x}_{0:T}^{(1:r)})$, we update the hyperparameters of the prior by maximizing the lower bound with respect to the hyperparameters. The variational lower bound simplifies to (see Ch.5 in [29] for details, note that the usage of Gaussian approximate posteriors ensures that this step is analogous to hyper parameter updating in a fully Gaussian LDS)

$$\log p(\mathbf{y}_{1:T}^{(1:r)}|\Phi) \geq -KL(\Phi) + c, \tag{8}$$

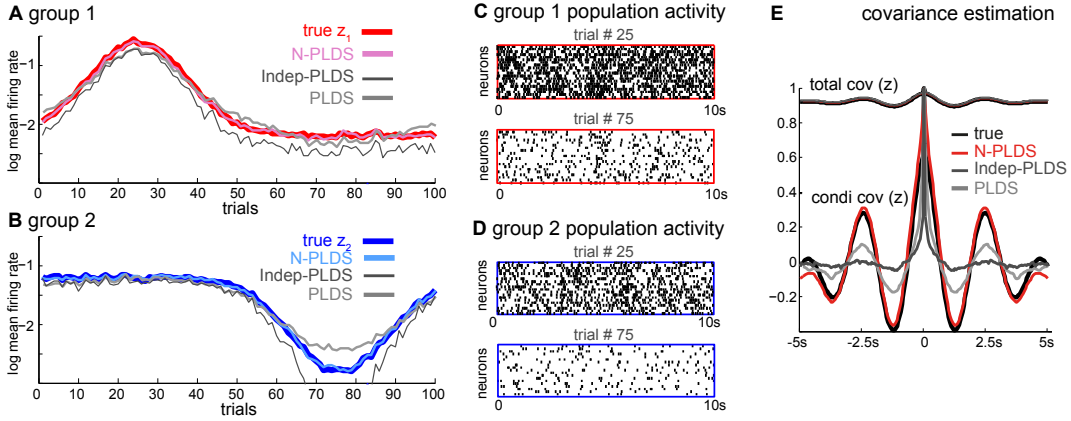

Figure 2: **Illustration of non-stationarity in firing rates (simulated data)**. **A, B** Spike rates of 40 neurons are influenced by two slowly varying firing rate modulators. The log mean firing rates of the two groups of neurons are $z_1$(red, group 1) and $z_2$(blue, group 2) across 100 trials. **C, D** Raster plots show the extreme cases, i.e. trials 25 and 75. The traces show the posterior mean of $z$ estimated by N-PLDS (light blue for $z_2$, light red for $z_1$), independent PLDSs (fit a PLDS to each trial data individually, dark gray), and PLDS (light gray). **E** Total and conditional (on each trial) covariance of recovered neural responses from each model (averaged across all neuron pairs, and then normalised for visualisation). The covariances recovered by our model (red) well match the true ones (black), while those by independent PLDSs (gray) and a single PLDS (light gray) do not.

where $c$ is a constant. Here, the KL divergence between the prior and posterior over parameters, denoted by $\mathcal{N}(\boldsymbol{\mu}_\Phi, \Sigma_\Phi)$ and $\mathcal{N}(\boldsymbol{\mu}, \Sigma)$, respectively, is given by

$$KL(\Phi) \;=\; -\tfrac{1}{2}\log|\Sigma_\Phi^{-1}\Sigma| + \tfrac{1}{2}\mathrm{Tr}\left[\Sigma_\Phi^{-1}\Sigma\right] + \tfrac{1}{2}(\boldsymbol{\mu}-\boldsymbol{\mu}_\Phi)^\top \Sigma_\Phi^{-1}(\boldsymbol{\mu}-\boldsymbol{\mu}_\Phi) + c, \qquad (9)$$

where the prior mean and covariance depend on the hyperparameters. We update the hyperparameters by taking the derivative of KL w.r.t. each hyper parameter. For the prior mean, the first derivative expression provides a closed-form update. For $\tau$ (time scale of inter-trial fluctuations in firing rates) and $\sigma^2$ (variance of inter-trial fluctuations), their derivative expressions do not provide a closed form update, in which case we compute the KL divergence on the grid defined in each hyperparameter space and choose the value that minimises KL.

**Predictive distributions for test data.** In our model, different trials are no longer considered to be independent, so we can predict parameters for held-out trials. Using the GP model on $\mathbf{h}$ and our approximations, we have Gaussian predictive distributions on $\mathbf{h}^*$ for test data $\mathcal{D}^*$ given training data $\mathcal{D}$:

$$p(\mathbf{h}^*|\mathcal{D}, \mathcal{D}^*) \;=\; \mathcal{N}(\mathbf{m_h} + K^*K^{-1}(\boldsymbol{\mu_h} - \mathbf{m_h}), \; K^{**} - K^*(K + H_\mathbf{h}^{-1})^{-1}K^{*\top}), \quad (10)$$

where $K$ is the prior covariance matrix on $\mathcal{D}$ and $K^{**}$ is on $\mathcal{D}^*$, and $K^*$ is their prior cross-covariance as introduced in Ch.2 of [28], and the negative Hessian $H_\mathbf{h}$ is defined as

$$H_\mathbf{h} = -\frac{\partial^2}{\partial^2 \mathbf{h}^{(1:h)}} \sum_{i=1}^{r}[\int d\mathbf{x}_{0:T}^{(i)} q(\mathbf{x}_{0:T}^{(i)}) \sum_{t=1}^{T} \log p(\mathbf{y}_t^{(i)}|\mathbf{x}_t^{(i)}, \hat{\mathbf{c}}, \hat{\mathbf{d}}, \mathbf{h}^{(i)})]. \qquad (11)$$

In the applications to simulated and neurophysiological data described in the following, we used this approach to predict the properties of neural dynamics on held-out trials.

## 4 Applications

**Simulated data:** We first illustrate the performance of N-PLDS on a simulated population recording from 40 neurons consisting of 100 trials of length $T = 200$ time steps each. We used a 4-dimensional latent state and assumed that the population consisted of two homogeneous sub-populations of size 20 each, with one modulatory input controlling rate fluctuations in each group (See Fig. 2 A). In addition, we assumed that for half of each trial, there was a time-varying stimulus ('drifting grating'), represented by a 3-dimensional vector which consisted of the sine and cosine

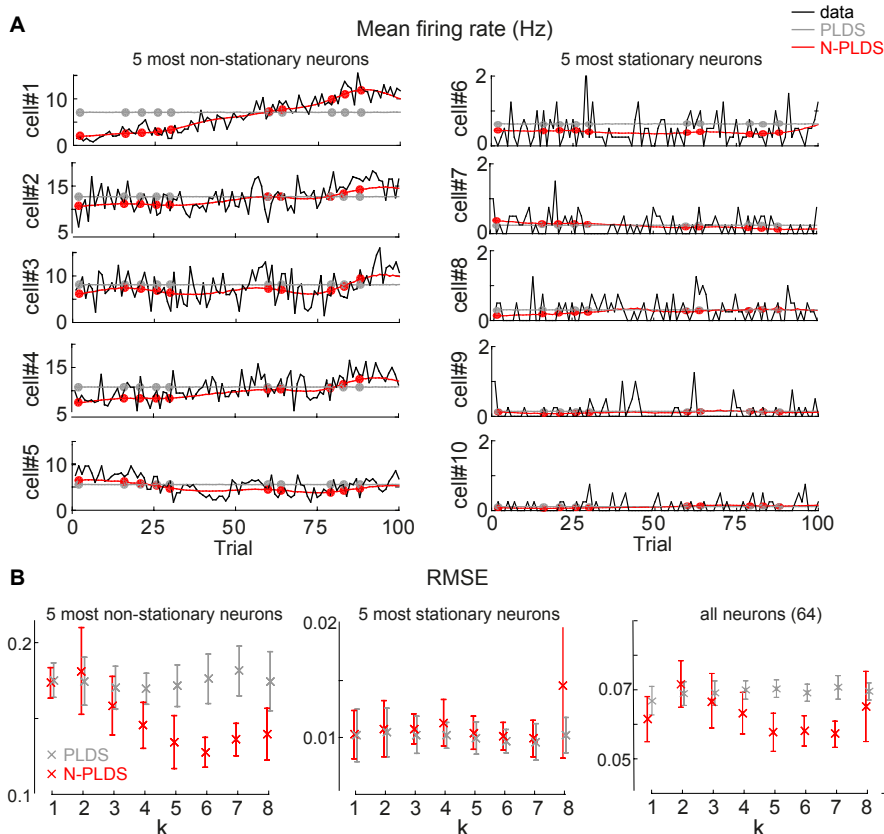

Figure 3: **Non-stationary firing rates in a population of V1 neurons. A**: Mean firing rates of neurons (black trace) across trials. Left: The 5 most non-stationary neurons. Right: The 5 most stationary neurons. The fitted (solid line) and the predicted (circles) mean firing rates are also shown for N-PLDS (in red) and PLDS (in gray). **B** Left: The RMSE in predicting single neuron firing rates across 5 most non-stationary neurons for varying latent dimensionalities $k$, where N-PLDS achieves significantly lower RMSE. Middle: RMSE for the 5 most stationary neurons, where there is no difference between two methods (apart from an outlier at k=8). Right: RMSE for the all $64$ neurons.

of the time-varying phase of the stimulus (frequency $0.4$ Hz) as well as an additional binary term which indicated whether the stimulus was active.

We fit N-PLDS to the data, and found that it successfully captures the non-stationarity in (log) mean firing rates, defined by $\mathbf{z} = C(\mathbf{x} + \mathbf{h}) + \mathbf{d}$, as shown in Fig. 2, and recovers the total and trial-conditioned covariances (the across-trial mean of the single-trial covariances of $\mathbf{z}$). For comparison, we also fit 100 separate PLDSs to the data from each trial, as well as a single PLDS to the entire data. The naive approach of fitting an individual PLDS to each trial can, in principle, follow the modulation. However, as each model is only fit to one trial, the parameter-estimates are very noisy since they are not sufficiently constrained by the data from each trial.

We note that a single PLDS with fixed parameters (as is conventionally used in neural data analysis) is able to track the modulations in firing rates in the posterior mean here– however, a single PLDS would not be able to extrapolate firing rates for unseen trials (as we will demonstrate in our analyses on neural data below). In addition, it will also fail to separate 'slow' and 'fast' modulations into different parameters. By comparing the total covariance of the data (averaged across neuron pairs) to the 'trial-conditioned' covariance (calculated by estimating the covariance on each trial individually, and averaging covariances) one can calculate how much of the cross-neuron co-variability can be explained by across-trials fluctuations in firing rates (see e.g., [17]). In this simulation shown in Fig. 2 (which illustrates an extreme case dominated by strong across-trial effects), the conditional covariance is much smaller than the full covariance.

**Neurophysiological data:** How big are non-stationarities in neural population recordings, and can our model successfully capture them? To address these questions, we analyzed a population recording from anaesthetized macaque primary visual cortex consisting of 64 neurons stimulated by sine grating stimuli. The details of data collection are described in [5], but our data-set also included units not used in the original study. We binned the spikes recorded during 100 trials of length 4s (stimulus was on for 2s) of the same orientation using 50ms bins, resulting in trials of length $T = 80$ bins. Analogously to the simulated dataset above, we parameterised the stimulus as a 3-dimensional vector of the sine and cosine with the same temporal frequency of the drifting grating, as well as an indicator that specifies whether there is a stimulus or not.

We used 10-fold cross validation to evaluate performance of the model, i.e. repeatedly divided the data into test data (10 trials) and training data (the remaining 90 trials). We fit the model on each training set, and using the estimated parameters from the training data, we made predictions on the modulator $\mathbf{h}$ on test data by using the mean of the predictive distribution over $\mathbf{h}$. We note that, in contrast to conventional applications of cross-validation which assume i.i.d. trials, our model here also takes into correlations in firing rates *across* trials– therefore, we had to keep the trial-indices in order to compute predictive distributions for test data using formulas in eq. 10. Using these parameters, we drew samples for spikes for the entire trials to compute the mean firing rates of each neuron at each trial. For comparison, we also fit a single PLDS to the data. As this model does not allow for across-trial modulations of firing rates, we simply kept the parameters estimated from the training data. For visualisation of results, we quantified the 'non-stationarity' of each neuron by first smoothing its firing rate across trials (using a kernel of size 10 trials), calculating the variance of the smoothed firing rate estimate, and displaying firing rates for the 5 most non-stationary neurons in the population (Fig. 3A, left) as well as 5 most stationary neurons (Fig. 3A, right). Importantly, the firing-rates were also correctly interpolated for held out trials (circles in Fig. 3A).

To evaluate whether the additional parameters in N-PLDS result in a superior model compared to conventional PLDS [13], we tested the model with different latent dimensionalities ranging from $k = 1$ to $k = 8$, and compared each model against a 'fixed' PLDS of matched dimensionality (Fig. 3B). We estimated predicted firing rates on held out trials by sampling 1000 replicate trials from the predictive distribution for both models and compared the median (across samples) of the mean firing rates of each neuron to those of the data. The shown RMSE values are the errors of predicted firing rate (in Hz) per neuron per held out trial (population mean across all neurons and trials is 4.54 Hz). We found that N-PLDS outperformed PLDS provided that we had sufficiently many latent states, at least $k > 3$. For large latent dimensionalities ($k > 8$) performance degraded again, which could be a consequence of overfitting. Furthermore, we show that for non-stationary neurons there is a large gain in predictive power (Fig. 3B, left), whereas for stationary neurons PLDS and N-PLDS have similar prediction accuracy (Fig. 3B, middle). The RMSE on firing rates for all neurons (Fig. 3B, right) suggests that our model correctly identified the fluctuation in firing rates.

We also wanted to gain insights into the temporal scale of the underlying non-stationarities. We first looked at the recovered time-scales $\tau$ of the latent modulators, and found them to be highly preserved across multiple training folds, and, importantly, across different values of the latent dimensionalities, consistently peaked near 10 trials (Fig. 4 A). We made sure that the peak near 10 trials is not merely a consequence of parameter initialization– parameters were initialised by fitting a Gaussian Process with a exponentiated quadratic one-dimensional kernel to each neuron's mean firing rate over trials individually, then taking the mean time-scale over neurons as the initial global time-scale for our kernel. The initial values were $8.12 \pm 0.01$, differing slightly between training sets. Similarly, we checked that the parameters of the final model (after 30 iterations of Bayesian Laplace propagation), were indeed superior to the initial values, by monitoring the prediction error on held-out trials. Furthermore, due to introducing a smooth change with the correct time scale in the latent space (e.g., the posterior mean of $\mathbf{h}$ across trials shown in Fig. 4B), we find that N-PLDS recovers more of the time-lagged covariance of neurons compared to the fixed PLDS model (Fig. 4C).

## 5 Discussion

Non-stationarities are ubiquitous in neural data: Slow modulations in firing properties can result from diverse processes such as plasticity and learning, fluctuations in arousal, cortical reorganisation after injury as well as development and aging. In addition, non-stationarities in neural data can also be a consequence of experimental artifacts, and can be caused by fluctuations in anaesthesia level,

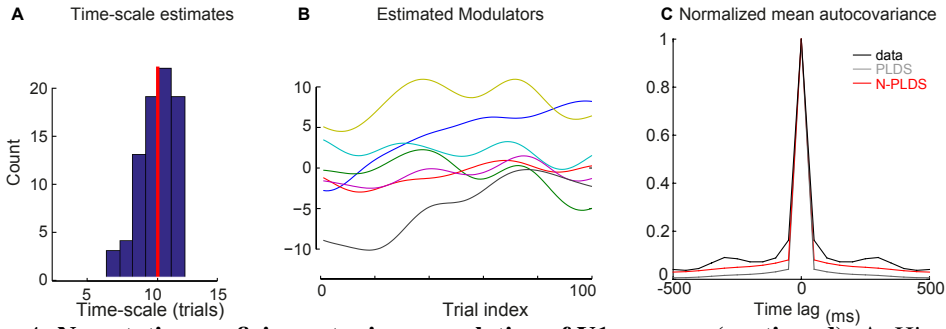

Figure 4: **Non-stationary firing rates in a population of V1 neurons (continued)**. **A**: Histogram of time-constants across different latent dimensionalities and training sets. Mean at $10.4$ is indicated by the vertical red line. **B**: Estimated 7-dimensional modulator (the posterior mean of $\mathbf{h}$). The modulator with an estimated length scale of approximately 10 trials is smoothly varying across trials. **C**: Comparison of normalized mean auto-covariance across neurons.

stability of the physiological preparation or electrode drift. Whatever the origins of non-stationarities are, it is important to have statistical models which can identify them and disentangle their effects from correlations and dynamics on faster time-scales [16].

We here presented a hierarchical model for neural population dynamics in the presence of non-stationarity. Specifically, we concentrated on a variant of this model which focuses on non-stationarity in firing rates. Recent experimental studies have shown that slow fluctuations in neural excitability which have a multiplicative effect on neural firing rates are a dominant source of noise correlations in anaesthetized visual cortex [17, 5, 24]. Because of the exponential spiking nonlinearity employed in our model, the latent additive fluctuations in the modulator-variables also have a multiplicative effect on firing rates. Applied to a data-set of neurophysiological recordings, we demonstrated that this modelling approach can successfully capture non-stationarities in neurophysiological recordings from primary visual cortex.

In our model, both neural dynamics and latent modulators are mediated by the same low-dimensional subspace (parameterised by $C$). We note, however, that this assumption does not imply that neurons with strong short-term correlations will also have strong long-term correlations, as different dimensions of this subspace (as long as it is chosen big enough) could be occupied by short and long term correlations, respectively. In our applications to neural data, we found that the latent state had to be at least three-dimensional for the non-stationary model to outperform a stationary dynamics model, and it might be the case that at least three dimensions are necessary to capture both fast and slow correlations. It is an open question of how correlations on fast and slow timescales are related [17], and the techniques presented have the potential to be of use for mapping out their relationships.

There are limitations to the current study: (1) We did not address the question of how to select amongst multiple different models which could be used to model neural non-stationarity for a given dataset; (2) we did not present numerical techniques for how to scale up the current algorithm for larger trial numbers (e.g., using low-rank approximations to the covariance matrix) or large neural populations; and (3) we did not address the question of how to overcome the slow convergence properties of GP kernel parameter estimation [34]. (4) While Laplace propagation is flexible, it is an approximate inference technique, and the quality of its approximations might vary for different models of tasks. We believe that extending our method to address these questions provides an exciting direction for future research, and will result in a powerful set of statistical methods for investigating how neural systems operate in the presence of non-stationarity.

### Acknowledgments

We thank Alexander Ecker and the lab of Andreas Tolias for sharing their data with us [5] (see `http://toliaslab.org/publications/ecker-et-al-2014/`), and for allowing us to use it in this publication, as well as Maneesh Sahani and Alexander Ecker for valuable comments. This work was funded by the Gatsby Charitable Foundation (MP and GB) and the German Federal Ministry of Education and Research (MP and JHM) through BMBF; FKZ:01GQ1002 (Bernstein Center Tübingen). Code available at `http://www.mackelab.org/code`.

## Footnotes

[1]A stochastic process is strict-sense stationary if its joint distribution over any two time-points $t$ and $s$ only depends on the elapsed time $t - s$.

[2]A second variant of the model, in which the dynamics matrix determining the spatio-temporal correlations in the population varies across trials, is described in the Appendix.

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
