[Supplementary Material · Appendix.pdf]



# Appendix

This appendix describes the Bayesian Laplace propagation algorithm we derived for the two variants of the hierarchical model we presented in the main text. First, we describe Model I for capturing nonstationarity in firing rates, and then we move to Model II for capturing nonstationarity in neural dynamics.

**Figure 1.** Schematic of hierarchical non-stationary Poisson observation Latent Dynamical System (N-PLDS) models. **Model I** for capturing non-stationarity in mean firing rates. The parameter $\mathbf{h}$ slowly varies across trials and leads to fluctuations in mean firing rates. **Model II** for capturing non-stationarity in population dynamics. The dynamics matrix $A$ changes across trials, as controlled by the hyperparameters $\Phi$.

# Model I : nonstationarity in firing rates

## Basic setup

**Likelihood**: $\mathbf{y}_t \in \mathbb{R}^p$, $\mathbf{x}_t \in \mathbb{R}^k$, $C \in \mathbb{R}^{p \times k}$

$$p(\mathbf{y}_t|\mathbf{x}_t, C, \mathbf{h}^{(i)}) = \text{Poiss}(\mathbf{y}_t|\exp(C(\mathbf{x}_t + \mathbf{h}^{(i)}) + \mathbf{d})),$$

where $\mathbf{h}^{(i)} \in \mathbb{R}^k$ is a vector of latent variables that capture nonstationarity in firing rates across recordings $i = \{1, \cdots, r\}$.

**Latent dynamics**: $A \in \mathbb{R}^{k \times k}$ and $B \in \mathbb{R}^{k \times d}$

$$p(\mathbf{x}_t|\mathbf{x}_{t-1}, A) = \mathcal{N}(\mathbf{x}_t|A\mathbf{x}_{t-1} + B\mathbf{u}_t, I).$$

Parameters in this model: $\Theta = \{A, B, C, \mathbf{h}^{(1:r)}\}$. For simplicity, we set $\mathbf{d}$ to its ML estimate. Vectorized notations: $\mathbf{a} = \text{vec}(A^\top) \in \mathbb{R}^{k^2}$, $\mathbf{b} = \text{vec}(B^\top) \in \mathbb{R}^{kd}$, and $\mathbf{c} = \text{vec}(C^\top) \in \mathbb{R}^{pk}$.

**Priors**:

$$p(\mathbf{a}|\alpha) = \mathcal{N}(\mathbf{a}|\mathbf{0}, \alpha^{-1}\mathbf{I}), \quad \mathbf{p}(\mathbf{b}|\beta) = \mathcal{N}(\mathbf{b}|\mathbf{0}, \beta^{-1}\mathbf{I}), \tag{1}$$

For $\mathbf{h}^{(i)}$, we assume slowly varying dynamics across recordings

$$\mathbf{h}^{(i)} \quad \sim \quad \mathcal{GP}(\mathbf{m_h}, K(i,j)) \tag{2}$$

where we denote the (vector) mean and (matrix) covariance functions by $K(i,j)$, respectively, where the $(i,j)$th block of the covariance matrix is given by

$$K(i,j) \quad = \quad (\sigma^2 + \epsilon\delta_{i,j})\exp\left(-\frac{1}{2\tau^2}(i-j)^2\right)I_k \tag{3}$$

The hyperparameters in total are $\Phi = \{\mathbf{m_h}, \alpha, \beta, \sigma^2, \tau^2\}$.

# Variational lower bound

The marginal likelihood of the observations is lower bounded by

$$\log p(\mathbf{y}_{1:T}^{(1:r)}) \geq \int d\theta \, d\mathbf{x}_{1:T}^{(1:r)} \, q(\theta, \mathbf{x}_{1:T}^{(1:r)}) \, \log \frac{p(\theta, \mathbf{x}_{1:T}^{(1:r)}, \mathbf{y}_{1:T}^{(1:r)})}{q(\theta, \mathbf{x}_{1:T}^{(1:r)})}, \tag{4}$$

where the approximate posterior factor is

$$q(\theta, \mathbf{x}_{1:T}^{(1:r)}) = q_\theta(\theta) \prod_{i=1}^{r} q_\mathbf{x}(\mathbf{x}_{0:T}^{(i)}), \tag{5}$$

$$\text{where} \quad q_\theta(\theta) = q_{\mathbf{a},\mathbf{b}}(\mathbf{a}, \mathbf{b}) \, q_{\mathbf{c},\mathbf{h}}(\mathbf{c}, \mathbf{h}^{(1:r)}). \tag{6}$$

For simplicity, we further assume

$$q_{\mathbf{c},\mathbf{h}}(\mathbf{c}, \mathbf{h}^{(1:r)}) = q_{\mathbf{h}|\mathbf{c}}(\mathbf{h}^{(1:r)}|\mathbf{c})q(\mathbf{c}), \tag{7}$$

$$= q_{\mathbf{h}|\mathbf{c}}(\mathbf{h}^{(1:r)}|\mathbf{c})\delta(\mathbf{c} - \hat{\mathbf{c}}), \tag{8}$$

$$= q_{\mathbf{h}|\hat{\mathbf{c}}}(\mathbf{h}^{(1:r)}|\hat{\mathbf{c}}), \tag{9}$$

where $\hat{\mathbf{c}}$ is maximum likelihood estimate of $\mathbf{c}$.

# Bayesian Laplace propagation

## Posterior over parameters

We compute $q_\theta(\theta)$ by integrating out latent variables from the total log joint distribution:

$$\log q_\theta(\theta) = \mathbb{E}_{q_\mathbf{x}(\mathbf{x}_{1:T}^{(1:r)})}\left[\log p(\mathbf{x}_{0:T}^{(1:r)}, \mathbf{y}_{1:T}^{(1:r)}, \theta)\right] + const, \tag{10}$$

$$= \mathbb{E}_{q_\mathbf{x}(\mathbf{x}_{0:T}^{(1:r)})}\left[\log p(\mathbf{y}_{1:T}^{(1:r)}|\mathbf{x}_{0:T}^{(1:r)}, \theta) + \log p(\mathbf{x}_{0:T}^{(1:r)}|\theta) + \log p(\theta)\right] + const,$$

$$= \sum_{i=1}^{r}[\mathbb{E}_{q_\mathbf{x}(\mathbf{x}_{0:T}^{(i)})}(\sum_{t=1}^{T}(\log p(\mathbf{y}_t^{(i)}|\mathbf{x}_t^{(i)}, \mathbf{c}, \mathbf{h}^{(i)}) + \log p(\mathbf{x}_t^{(i)}|\mathbf{x}_{t-1}^{(i)}, \mathbf{u}_t, \mathbf{a}, \mathbf{b}))] + \log p(\mathbf{a}|\alpha) + \log p(\mathbf{b}|\beta)$$

$$+ \log p(\mathbf{h}^{(1:r)}|0, K) + const.$$

Note that we assume the inputs $\mathbf{u}$ are the same across recordings. (so we don't put the recording index $i$ on $\mathbf{u}_t$).

### 1. approximate posterior over $\mathbf{a}, \mathbf{b}$

We can compute $q_{\mathbf{a},\mathbf{b}}(\mathbf{a}, \mathbf{b})$ by extracting all the terms in $\log p(\mathbf{x}_{0:T}, \mathbf{y}_{1:T}, \theta)$ that depend on $\mathbf{a}, \mathbf{b}$ and then taking the expectation of the terms w.r.t. $q_\mathbf{x}(\mathbf{x}_{0:T})$:

$$\log q_{\mathbf{a},\mathbf{b}}(\mathbf{a}, \mathbf{b}) = \sum_{i=1}^{r} \mathbb{E}_{q_\mathbf{x}(\mathbf{x}_{0:T}^{(i)})}\left[\sum_{t=1}^{T} \log p(\mathbf{x}_t^{(i)}|\mathbf{x}_{t-1}^{(i)}, \mathbf{u}_t, \mathbf{a}, \mathbf{b})\right] + \log p(\mathbf{a}|\alpha) + \log p(\mathbf{b}|\beta) + const,$$

$$= -\frac{1}{2}\sum_{i=1}^{r}\mathbb{E}_{q_\mathbf{x}(\mathbf{x}_{0:T}^{(i)})}\left[\sum_{t=1}^{T}(\mathbf{x}_t^{(i)} - A\mathbf{x}_{t-1}^{(i)} - B\mathbf{u}_t)^\top(\mathbf{x}_t^{(i)} - A\mathbf{x}_{t-1}^{(i)} - B\mathbf{u}_t)\right] - \frac{\alpha}{2}\mathbf{a}^\top\mathbf{a} - \frac{\beta}{2}\mathbf{b}^\top\mathbf{b} + const, \tag{11}$$

$$= -\frac{1}{2}\sum_{i=1}^{r}\left[\mathbf{a}^\top(\frac{\alpha}{r}I + W_{A^{(i)}}^{bd})\mathbf{a} - 2\mathbf{a}^\top(\text{vec}(S_{A^{(i)}}) - G_{A^{(i)}}^{bd}\mathbf{b}) + \mathbf{b}^\top(\frac{\beta}{r}I + \ddot{U}^{bd})\mathbf{b} - 2\mathbf{b}^\top\text{vec}(\tilde{M}^{(i)})\right] + const,$$

where $W_{A^{(i)}}^{bd} = I_r \otimes W_{A^{(i)}}$, $G_{A^{(i)}}^{bd} = I_r \otimes G_{A^{(i)}}$, $\ddot{U}^{bd} = I_r \otimes \ddot{U}$, and the sufficient statistics are denoted by

$$W_{A^{(i)}} = \sum_{t=1}^T < \mathbf{x}_{t-1}^{(i)} \mathbf{x}_{t-1}^{(i)\top} >, \quad S_{A^{(i)}} = \sum_{t=1}^T < \mathbf{x}_{t-1}^{(i)} \mathbf{x}_t^{(i)\top} >, \qquad G_{A^{(i)}} = \sum_{t=1}^T < \mathbf{x}_{t-1}^{(i)} > \mathbf{u}_t^\top,$$

$$\ddot{U} = \sum_{t=1}^T \mathbf{u}_t \mathbf{u}_t^\top, \qquad\qquad \tilde{M}^{(i)} = \sum_{t=1}^T \mathbf{u}_t < \mathbf{x}_t^{(i)} >^\top \tag{12}$$

Using new notations $\mathbf{W} = \sum_{i=1}^r W_{A^{(i)}}^{bd}$, $\mathbf{s} = \sum_{i=1}^r \text{vec}(S_{A^{(i)}})$, $\mathbf{G} = \sum_{i=1}^r G_{A^{(i)}}^{bd}$, $\mathbf{m} = \sum_{i=1}^r \text{vec}(\tilde{M}^{(i)})$, we rewrite eq. 11 , whose derivative expressions are given by

$$\log q_{\mathbf{a},\mathbf{b}}(\mathbf{a},\mathbf{b}) = -\tfrac{1}{2}\left[\mathbf{a}^\top(\alpha I + \mathbf{W})\mathbf{a} - 2\mathbf{a}^\top(\mathbf{s} - \mathbf{G}\mathbf{b}) + \mathbf{b}^\top(\beta I + r\ddot{U}^{bd})\mathbf{b} - 2\mathbf{b}^\top\mathbf{m}\right], \tag{13}$$

$$H_{\mathbf{a}} = -\tfrac{\partial}{\partial \mathbf{a}\mathbf{a}^\top}\log q_{\mathbf{a},\mathbf{b}}(\mathbf{a},\mathbf{b}) = (\alpha I + \mathbf{W}), \tag{14}$$

$$H_{\mathbf{ab}} = -\tfrac{\partial}{\partial \mathbf{a}\mathbf{b}^\top}\log q_{\mathbf{a},\mathbf{b}}(\mathbf{a},\mathbf{b}) = \mathbf{G}, \tag{15}$$

$$H_{\mathbf{b}} = -\tfrac{\partial}{\partial \mathbf{b}\mathbf{b}^\top}\log q_{\mathbf{a},\mathbf{b}}(\mathbf{a},\mathbf{b}) = (\beta I + r\ddot{U}^{bd}), \tag{16}$$

$$\tfrac{\partial}{\partial \mathbf{a}}\log q_{\mathbf{a},\mathbf{b}}(\mathbf{a},\mathbf{b}) = -(\alpha I + \mathbf{W})\mathbf{a} + (\mathbf{s} - \mathbf{G}\mathbf{b}) = -H_{\mathbf{a}}\mathbf{a} + (\mathbf{s} - \mathbf{G}\mathbf{b}), \tag{17}$$

$$\tfrac{\partial}{\partial \mathbf{b}}\log q_{\mathbf{a},\mathbf{b}}(\mathbf{a},\mathbf{b}) = -\mathbf{G}^\top\mathbf{a} - (\beta I + r\ddot{U}^{bd})\mathbf{b} + \mathbf{m} = -\mathbf{G}^\top\mathbf{a} - H_{\mathbf{b}}\mathbf{b} + \mathbf{m}. \tag{18}$$

Using Schur complement, we obtain the covariance of $q(\mathbf{a},\mathbf{b})$

$$\Sigma_{\mathbf{a}} = (H_{\mathbf{a}} - H_{\mathbf{ab}}H_{\mathbf{b}}^{-1}H_{\mathbf{ab}}^\top)^{-1}, \tag{19}$$

$$\Sigma_{\mathbf{b}} = (H_{\mathbf{b}} - H_{\mathbf{ab}}^\top H_{\mathbf{a}}^{-1}H_{\mathbf{ab}})^{-1}, \tag{20}$$

$$\Sigma_{\mathbf{ab}} = -\Sigma_{\mathbf{a}}H_{\mathbf{ab}}H_{\mathbf{b}}^{-1}, \tag{21}$$

and the mean of $q(\mathbf{a},\mathbf{b})$,

$$\boldsymbol{\mu}_{\mathbf{b}} = \Sigma_{\mathbf{b}}(\mathbf{m} - H_{\mathbf{ab}}^\top H_{\mathbf{a}}^{-1}\mathbf{s}), \tag{22}$$

$$\boldsymbol{\mu}_{\mathbf{a}} = \Sigma_{\mathbf{a}}(\mathbf{s} - H_{\mathbf{ab}}H_{\mathbf{b}}^{-1}\mathbf{m}). \tag{23}$$

## 2. Computing $q_{\mathbf{h}|\hat{\mathbf{c}}}(\mathbf{h}^{(1:r)}|\hat{\mathbf{c}})$

Assuming we have the maximum likelihood estimate of $\mathbf{c}$, we write down all the terms in $\log p(\mathbf{x}_{0:T}, \mathbf{y}_{1:T}, \theta)$ that depend on $\mathbf{h}^{(1:r)}$:

$$\log q_{\mathbf{h}|\hat{\mathbf{c}}}(\mathbf{h}^{(1:r)}|\hat{\mathbf{c}}) = \sum_{i=1}^r \mathbb{E}_{q_{\mathbf{x}}(\mathbf{x}_{1:T}^{(i)})}\left[\sum_{t=1}^T \log p(\mathbf{y}_t^{(i)}|\mathbf{x}_t^{(i)},\hat{\mathbf{c}},\mathbf{h}^{(i)})\right] + \log p(\mathbf{h}^{(1:r)}|0,K),$$

$$= \sum_{i=1}^r \mathbb{E}_{q_{\mathbf{x}}(\mathbf{x}_{1:T}^{(i)})}\left[\sum_{t=1}^T (\mathbf{y}_t^{(i)\top}(\hat{C}(\mathbf{x}_t^{(i)} + \mathbf{h}^{(i)}) + \mathbf{d}) - \mathbf{1}^\top \exp(\hat{C}(\mathbf{x}_t^{(i)} + \mathbf{h}^{(i)}) + \mathbf{d}))\right] - \tfrac{1}{2}\mathbf{h}^{(1:r)\top}K^{-1}\mathbf{h}^{(1:r)} + const,$$

$$= \sum_{i=1}^r \left[\hat{C}S_{C^{(i)}} + \sum_{t=1}^T \mathbf{y}_t^{(i)\top}(\hat{C}\mathbf{h}^{(i)} + \mathbf{d}) - \mathbb{E}_{q_{\mathbf{x}}(\mathbf{x}_{1:T}^{(i)})}\mathbf{1}^\top \exp(\hat{C}(\mathbf{x}_t^{(i)} + \mathbf{h}^{(i)}) + \mathbf{d}))\right] - \tfrac{1}{2}\mathbf{h}^{(1:r)\top}K^{-1}\mathbf{h}^{(1:r)} + const,$$

$$\tag{24}$$

where each row of $\hat{C}$ is denoted by $\hat{\mathbf{c}}_s$ and the sufficient statistic is denoted by

$$S_{C^{(i)}} = \sum_{t=1}^T < \mathbf{x}_t^{(i)} > \mathbf{y}_t^{(i)\top} \tag{25}$$

Assuming the approximate posterior over latent variables is multivariate Gaussian with marginals $q(\mathbf{x}_t) = \mathcal{N}(\mathbf{x}_t|\boldsymbol{\omega}_t, \Upsilon_t)$, the expectation of the exponential term above is given by

$$\mathbb{E}_{q_{\mathbf{x}}(\mathbf{x}_{1:T}^{(i)})}\{\sum_{t=1}^{T} \exp(\hat{\mathbf{c}}_s^\top \mathbf{x}_t^{(i)})\} = \int d\mathbf{x}_{1:T}^{(i)} q_{\mathbf{x}}(\mathbf{x}_{1:T}^{(i)}) \exp(\hat{\mathbf{c}}_s^\top \mathbf{x}_1^{(i)} + \cdots + \hat{\mathbf{c}}_s^\top \mathbf{x}_T^{(i)}),$$

$$= \sum_{t=1}^{T} \exp(\hat{\mathbf{c}}_s^\top \boldsymbol{\omega}_t^{(i)} + \tfrac{1}{2}\hat{\mathbf{c}}_s^\top \Upsilon_t^{(i)} \hat{\mathbf{c}}_s). \tag{26}$$

Therefore, the log joint distribution is given by

$$\log q_{\mathbf{h}|\hat{\mathbf{c}}}(\mathbf{h}^{(1:r)}|\hat{\mathbf{c}}) = \sum_{i=1}^{r}\left[\hat{C}S_{C^{(i)}} + \sum_{t=1}^{T}\left(\mathbf{y}_t^{(i)\top}(\hat{C}\mathbf{h}^{(i)} + \mathbf{d}) - \mathbf{1}^\top \exp(\hat{C}(\boldsymbol{\omega}_t^{(i)} + \mathbf{h}^{(i)}) + \tfrac{1}{2}\mathrm{diag}(\hat{C}\Upsilon_t^{(i)}\hat{C}^\top) + \mathbf{d})\right)\right] - \tfrac{1}{2}\mathbf{h}^{(1:r)\top}K^{-1}\mathbf{h}^{(1:r)},$$

$$= \mathbf{h}^{(1:r)\top}(\hat{C}^{bd\top}\mathbf{y}^{(1:r)}) - \mathbf{1}^\top(\exp(\hat{C}^{bd}\mathbf{h}^{(1:r)}) \circ \mathbf{g}^{(1:r)}) - \tfrac{1}{2}\mathbf{h}^{(1:r)\top}K^{-1}\mathbf{h}^{(1:r)}, \tag{27}$$

where $\hat{C}^{bd} = I_r \otimes \hat{C}$, $\mathbf{y}^{(1:r)} = [\sum_{t=1}^{T}\mathbf{y}_t^{(1)}, \cdots, \sum_{t=1}^{T}\mathbf{y}_t^{(r)}]^\top$, $\mathbf{g}^{(1:r)} = [\mathbf{g}^{(1)}, \cdots, \mathbf{g}^{(r)}]^\top$, where $\mathbf{g}^{(i)} = \sum_{t=1}^{T}\exp(\hat{C}\boldsymbol{\omega}_t^{(i)} + \tfrac{1}{2}\mathrm{diag}(\hat{C}\Upsilon_t^{(i)}\hat{C}^\top) + \mathbf{d})$.

We approximate the joint posterior $q_{\mathbf{h}|\hat{\mathbf{c}}}(\mathbf{h}^{(1:r)}|\hat{\mathbf{c}})$ as a Gaussian distribution from the derivatives w.r.t. $\mathbf{h}^{(1:r)}$:

$$q_{\mathbf{h}|\hat{\mathbf{c}}}(\mathbf{h}^{(1:r)}|\hat{\mathbf{c}}) = \mathcal{N}(\mathbf{h}^{(1:r)}|\boldsymbol{\mu}_{\mathbf{h}}, \Sigma_{\mathbf{h}}) \tag{28}$$

$$\Sigma_{\mathbf{h}}^{-1} = H_{\mathbf{h}} + K^{-1}, \quad \text{where } \mathbf{h}^{(1:r)} = \boldsymbol{\mu}_{\mathbf{h}}, \tag{29}$$

$$\boldsymbol{\mu}_{\mathbf{h}} = K\left[\hat{C}^{bd\top}\mathbf{y}^{(1:r)} - \hat{C}^{bd\top}(\exp(\hat{C}^{bd}\mathbf{h}^{(1:r)}) \circ \mathbf{g}^{(1:r)})\right], \quad \text{where } \mathbf{h}^{(1:r)} = \boldsymbol{\mu}_{\mathbf{h}} \tag{30}$$

where $H_{\mathbf{h}} = -\frac{\partial^2}{\partial^2 \mathbf{h}^{(1:h)}}\sum_{i=1}^{r}[\int d\mathbf{x}_{0:T}^{(i)} q(\mathbf{x}_{0:T}^{(i)})\sum_{t=1}^{T}\log p(\mathbf{y}_t^{(i)}|\mathbf{x}_t^{(i)}, \hat{\mathbf{c}}, \hat{\mathbf{d}}, \mathbf{h}^{(i)})] = \hat{C}^{bd\top}\mathrm{diag}\left[\exp(\hat{C}^{bd}\mathbf{h}^{(1:r)}) \circ \mathbf{g}^{(1:r)}\right]\hat{C}^{bd}$.

## 3. Computing the ML estimate of $\hat{\mathbf{c}}$

We set $C$ to the ML estimate $\hat{C}$, which is obtained by

$$\hat{\mathbf{c}} = \arg\max_{\mathbf{c}}\sum_{i=1}^{r}\mathbb{E}_{q_{\mathbf{x}}(\mathbf{x}_{1:T}^{(i)})}\left[\sum_{t=1}^{T}\log p(\mathbf{y}_t^{(i)}|\mathbf{x}_t^{(i)}, \mathbf{c}, \mathbf{h}^{(i)})\right], \tag{31}$$

whose first derivatives w.r.t. $C$ is given by :

$$\sum_{i=1}^{r}\left[S_{C^{(i)}}^\top + \sum_{t=1}^{T}(\mathbf{y}_t\mathbf{h}^{(i)\top} - l^{(i)}(C)(\boldsymbol{\omega}_t^{(i)} + \mathbf{h}^{(i)})^\top - \mathrm{diag}(l^{(i)}(C))\Upsilon_t^{(i)}C\right] \tag{32}$$

where we fix $\mathbf{h}^{(i)}$ to its posterior mean $\boldsymbol{\mu}_{\mathbf{h}^{(i)}}$ and $l^{(i)}(C) = \exp(C^\top(\boldsymbol{\omega}_t^{(i)} + \mathbf{h}^{(i)}) + \tfrac{1}{2}\mathrm{diag}(C^\top\Upsilon_t^{(i)}C) + \mathbf{d})$.

## 4. ML estimate of d

$$\hat{\mathbf{d}} = \arg\max_{\mathbf{d}}\sum_{i=1}^{r}\left[\hat{C}S_{C^{(i)}} + \sum_{t=1}^{T}\left(\mathbf{y}_t^{(i)\top}(\hat{C}\mathbf{h}^{(i)} + \mathbf{d}) - \mathbf{1}^\top\exp(\hat{C}(\boldsymbol{\omega}_t^{(i)} + \mathbf{h}^{(i)}) + \tfrac{1}{2}\mathrm{diag}(\hat{C}\Upsilon_t^{(i)}\hat{C}^\top) + \mathbf{d})\right)\right], \tag{33}$$

$$= \log(\sum_{i=1}^{r}\sum_{t=1}^{T}\mathbf{y}_t^{(i)}) - \log(\sum_{i=1}^{r}\sum_{t=1}^{T}\exp(\hat{C}(\boldsymbol{\omega}_t^{(i)} + \mathbf{h}^{(i)}) + \tfrac{1}{2}\mathrm{diag}(\hat{C}\Upsilon_t^{(i)}\hat{C}^\top))). \tag{34}$$

## Posterior over latent variables

In VBE step, we compute $q_{\mathbf{x}}(\mathbf{x}_{0:qT})$ by

$$
\begin{aligned}
\sum_{i=1}^{r} \log q_{\mathbf{x}}(\mathbf{x}_{0:T}^{(i)}) &= \sum_{i=1}^{r} \mathbb{E}_{q_\theta(\theta)} \log p(\theta, \mathbf{x}_{0:T}^{(i)}, \mathbf{y}_{1:T}^{(i)}) + const, \\
&= \sum_{i=1}^{r} \left[ \mathbb{E}_{q_\theta(\theta)} \log p(\mathbf{x}_{0:T}^{(i)}, \mathbf{y}_{1:T}^{(i)}|\theta) - \log Z'_{(i)} \right],
\end{aligned}
\tag{35}
$$

where the normalization constant is given by

$$
Z'_{(i)} = \int d\mathbf{x}_{0:T}^{(i)} \exp \left( \mathbb{E}_{q_\theta(\theta)} \log p(\mathbf{x}_{0:T}^{(i)}, \mathbf{y}_{1:T}^{(i)}|\theta) \right).
\tag{36}
$$

The complete-data log likelihood in the $i$th recording is written as

$$
\log p(\mathbf{x}_{0:T}^{(i)}, \mathbf{y}_{1:T}^{(i)}|\theta) = \sum_{t=1}^{T} \{ \log p(\mathbf{y}_t^{(i)}|\mathbf{x}_t^{(i)}, C, \mathbf{d}, \mathbf{h}^{(i)}) + \log p(\mathbf{x}_t^{(i)}|\mathbf{x}_{t-1}^{(i)}, A, B, \mathbf{u}_t) \},
\tag{37}
$$

which tells us that the log posterior over latent variables is quadratic in each $\mathbf{x}_t$. This enables us to use the sequential update of the posterior over latent variables. We will also use the following sequential forward/backward algorithm for each recording in parallel. In the following, the recording index $i$ on $\mathbf{x}, \mathbf{y}$ is removed for notational cleanness.

### Forward filtering

We denote the posterior over the latent variables at each time $t$ by

$$
\begin{aligned}
\alpha(\mathbf{x}_t) &\propto \int d\mathbf{x}_{t-1} \alpha(\mathbf{x}_{t-1}) \exp \left[ < \log(p(\mathbf{x}_t|\mathbf{x}_{t-1}) p(\mathbf{y}_t|\mathbf{x}_t)) >_{q_\theta(\theta)} \right], \tag{38} \\
&\propto \exp(< \log p(\mathbf{y}_t|\mathbf{x}_t)) >_{q_\theta(\theta)} \left\{ \int d\mathbf{x}_{t-1} \alpha(\mathbf{x}_{t-1}) \exp \left( < \log(p(\mathbf{x}_t|\mathbf{x}_{t-1}) >_{q(\theta)} \right) \right\}. \tag{39}
\end{aligned}
$$

Assuming $\alpha(\mathbf{x}_{t-1}) = \mathcal{N}(\mathbf{x}_{t-1}|\boldsymbol{\mu}_{t-1}, \Sigma_{t-1})$, the integral is analytically tractable since the second part in the integrand is also quadratic in $\mathbf{x}_{t-1}$:

$$
\exp[-\tfrac{1}{2}(\mathbf{x}_{t-1}^\top < A^\top A > \mathbf{x}_{t-1} - 2\mathbf{x}_{t-1}^\top (< A >^\top \mathbf{x}_t - < A^\top B > \mathbf{u}_t) + \mathbf{x}_t^\top \mathbf{x}_t - 2\mathbf{x}_t^\top < B > \mathbf{u}_t + \mathbf{u}_t^\top < B^\top B > \mathbf{u}_t)].
$$

The integrand is summarised as

$$
\begin{aligned}
\alpha(\mathbf{x}_{t-1}) \exp \left( < \log(p(\mathbf{x}_t|\mathbf{x}_{t-1}) >_{q(\theta)} \right) &= Z\mathcal{N}(\mathbf{x}_{t-1}|\boldsymbol{\mu}_{t-1}^*, \Sigma_{t-1}^*), \tag{40} \\
\Sigma_{t-1}^{*-1} &= \Sigma_{t-1}^{-1} + < A^\top A >, \tag{41} \\
\boldsymbol{\mu}_{t-1}^* &= \Sigma_{t-1}^*(\Sigma_{t-1}^{-1}\mu_{t-1} + < A >^\top \mathbf{x}_t - < A^\top B > \mathbf{u}_t), \tag{42}
\end{aligned}
$$

and the remaining term $Z$ is given by:

$$
Z = \exp[-\tfrac{1}{2}(\mathbf{x}_t^\top \mathbf{x}_t - 2\mathbf{x}_t^\top < B > \mathbf{u}_t + \mathbf{u}_t^\top < B^\top B > \mathbf{u}_t) + \tfrac{1}{2}\boldsymbol{\mu}_{t-1}^{*\top}\Sigma_{t-1}^{*-1}\boldsymbol{\mu}_{t-1}^*],
\tag{43}
$$

where

$$
\begin{aligned}
\tfrac{1}{2}\boldsymbol{\mu}_{t-1}^{*\top}\Sigma_{t-1}^{*-1}\boldsymbol{\mu}_{t-1}^* &= \tfrac{1}{2}(\Sigma_{t-1}^{-1}\mu_{t-1} + < A >^\top \mathbf{x}_t - < A^\top B > \mathbf{u}_t)^\top \Sigma_{t-1}^*(\Sigma_{t-1}^{-1}\mu_{t-1} + < A >^\top \mathbf{x}_t - < A^\top B > \mathbf{u}_t), \\
&= \tfrac{1}{2}(\mathbf{x}_t^\top < A > \Sigma_{t-1}^* < A >^\top \mathbf{x}_t + 2\mathbf{x}_t^\top < A > (\Sigma_{t-1}^*\Sigma_{t-1}^{-1}\mu_{t-1} - \Sigma_{t-1}^* < A^\top B > \mathbf{u}_t) + \\
&\quad \mu_{t-1}^\top \Sigma_{t-1}^{-1}\Sigma_{t-1}^*\Sigma_{t-1}^{-1}\mu_{t-1} - 2\mu_{t-1}^\top \Sigma_{t-1}^{-1}\Sigma_{t-1}^* < A^\top B > \mathbf{u}_t + \mathbf{u}_t^\top < A^\top B >^\top \Sigma_{t-1}^* < A^\top B > \mathbf{u}_t).
\end{aligned}
$$

Therefore, $Z$ is proportional to a Gaussian in $\mathbf{x}_t$ :

$$
\begin{aligned}
Z &\propto \mathcal{N}(\mathbf{x}_t|\tilde{\boldsymbol{\mu}}_t, \tilde{\Sigma}_t), & (44)\\
\tilde{\Sigma}_t^{-1} &= I - <A> \Sigma_{t-1}^* <A>^\top, & (45)\\
\tilde{\boldsymbol{\mu}}_t &= \tilde{\Sigma}_t(<B>\mathbf{u}_t + <A>\Sigma_{t-1}^*\Sigma_{t-1}^{-1}\boldsymbol{\mu}_{t-1} - <A>\Sigma_{t-1}^* <A^\top B>\mathbf{u}_t), & (46)
\end{aligned}
$$

We approximate the forward message as a Gaussian in $\mathbf{x}_t$ using the first and second derivatives w.r.t. $\mathbf{x}_t$

$$
\alpha(\mathbf{x}_t) \propto \exp(<\log p(\mathbf{y}_t|\mathbf{x}_t)>_{q_\theta(\theta)})\mathcal{N}(\mathbf{x}_t|\tilde{\boldsymbol{\mu}}_t, \tilde{\Sigma}_t). \tag{47}
$$

where

$$
\begin{aligned}
<\log p(\mathbf{y}_t|\mathbf{x}_t)>_{q_\theta(\theta)} &= \int \log p(\mathbf{y}_t|\mathbf{x}_t, \hat{C}, \mathbf{d}, \mathbf{h}^{(i)})\mathcal{N}(\mathbf{h}^{(i)}|\boldsymbol{\mu}_{\mathbf{h}^{(i)}}, \Sigma_{\mathbf{h}^{(i)}})d\mathbf{h}^{(i)}, & (48)\\
&= \int \left[\mathbf{y}_t^\top(\hat{C}(\mathbf{x}_t + \mathbf{h}^{(i)}) + \mathbf{d}) - \mathbf{1}^\top\exp(\hat{C}(\mathbf{x}_t + \mathbf{h}^{(i)}) + \mathbf{d})\right]\mathcal{N}(\mathbf{h}^{(i)}|\boldsymbol{\mu}_{\mathbf{h}^{(i)}}, \Sigma_{\mathbf{h}^{(i)}})d\mathbf{h}^{(i)}, & (49)\\
&= \sum_{s=1}^p \int \left[(\mathbf{y}_t^\top\mathbf{e}_s)(\mathbf{x}_t^\top\hat{\mathbf{c}}_s) - \exp(\mathbf{x}_t^\top\hat{\mathbf{c}}_s + \mathbf{h}^{(i)\top}\hat{\mathbf{c}}_s + \mathbf{d}_s)\right]\mathcal{N}(\mathbf{h}^{(i)}|\boldsymbol{\mu}_{\mathbf{h}^{(i)}}, \Sigma_{\mathbf{h}^{(i)}})d\mathbf{h}^{(i)}, & (50)\\
&= \sum_{s=1}^p \left[(\mathbf{y}_t^\top\mathbf{e}_s)(\mathbf{x}_t^\top\hat{\mathbf{c}}_s) - \exp(\mathbf{x}_t^\top\hat{\mathbf{c}}_s + \hat{\mathbf{c}}_s^\top\boldsymbol{\mu}_{\mathbf{h}^{(i)}} + \frac{1}{2}\hat{\mathbf{c}}_s^\top\Sigma_{\mathbf{h}^{(i)}}\hat{\mathbf{c}}_s + \mathbf{d}_s)\right] & (51)
\end{aligned}
$$

The forward message at time $t$ is approximately

$$
\begin{aligned}
\alpha(\mathbf{x}_t) &\approx \mathcal{N}(\mathbf{x}_t|\boldsymbol{\mu}_t, \Sigma_t), & (52)\\
\boldsymbol{\mu}_t &= \tilde{\boldsymbol{\mu}}_t + \tilde{\Sigma}_t\sum_{s=1}^p \left[\mathbf{y}_t^T\mathbf{e}_s - \exp(\mathbf{x}_t^\top\hat{\mathbf{c}}_s + \hat{\mathbf{c}}_s^\top\boldsymbol{\mu}_{\mathbf{h}^{(i)}} + \frac{1}{2}\hat{\mathbf{c}}_s^\top\Sigma_{\mathbf{h}^{(i)}}\hat{\mathbf{c}}_s + \mathbf{d}_s)\right]\hat{\mathbf{c}}_s, \quad \text{where } \mathbf{x}_t = \boldsymbol{\mu}_t, & (53)\\
\Sigma_t^{-1} &= \tilde{\Sigma}_t^{-1} + \sum_{s=1}^p \exp(\mathbf{x}_t^\top\hat{\mathbf{c}}_s + \hat{\mathbf{c}}_s^\top\boldsymbol{\mu}_{\mathbf{h}^{(i)}} + \frac{1}{2}\hat{\mathbf{c}}_s^\top\Sigma_{\mathbf{h}^{(i)}}\hat{\mathbf{c}}_s + \mathbf{d}_s)\hat{\mathbf{c}}_s\hat{\mathbf{c}}_s^\top, \quad \text{where } \mathbf{x}_t = \boldsymbol{\mu}_t. & (54)
\end{aligned}
$$

**Backward smoothing**

We denote the backward message at each time $t$ by

$$
\beta(\mathbf{x}_t) = p(\mathbf{y}_{t+1:T}|\mathbf{x}_t) = \mathcal{N}(\mathbf{x}_t|\boldsymbol{\eta}_t, \Psi_t). \tag{55}
$$

We can obtain the recursion rules by considering $\beta(\mathbf{x}_{t-1})$

$$
\begin{aligned}
\beta(\mathbf{x}_{t-1}) &= \int d\mathbf{x}_t \beta(\mathbf{x}_t)\exp\left(<\log(p(\mathbf{x}_t|\mathbf{x}_{t-1})p(\mathbf{y}_t|\mathbf{x}_t))>_{q_\theta(\theta)}\right),\\
&= \int d\mathbf{x}_t \exp\left(<\log(p(\mathbf{x}_t|\mathbf{x}_{t-1})>_{q_\theta(\theta)}\right)\left[\beta(\mathbf{x}_t)\exp\left(<\log p(\mathbf{y}_t|\mathbf{x}_t))>_{q_\theta(\theta)}\right)\right],\\
&= \int d\mathbf{x}_t \exp\left(<\log(p(\mathbf{x}_t|\mathbf{x}_{t-1})>_{q_\theta(\theta)}\right)\mathcal{N}(\mathbf{x}_t|\tilde{\boldsymbol{\eta}}_t, \tilde{\Psi}_t), & (56)
\end{aligned}
$$

assuming $\beta(\mathbf{x}_T) = 1$. The Gaussian $p(\mathbf{x}_t) = \mathcal{N}(\mathbf{x}_t|\tilde{\boldsymbol{\eta}}_t, \tilde{\Psi}_t)$ is obtained by computing the first and second derivatives w.r.t. $\mathbf{x}_t$,

$$\tilde{\boldsymbol{\eta}}_t = \boldsymbol{\eta}_t + \Psi_t \sum_{s=1}^{p} \left[ \mathbf{y}_t^T \mathbf{e}_s - \exp(\mathbf{x}_t^\top \hat{\mathbf{c}}_s + \hat{\mathbf{c}}_s^\top \boldsymbol{\mu}_{\mathbf{h}^{(i)}} + \tfrac{1}{2}\hat{\mathbf{c}}_s^\top \Sigma_{\mathbf{h}^{(i)}} \hat{\mathbf{c}}_s + \mathbf{d}_s) \right] \hat{\mathbf{c}}_s, \quad \text{where } \mathbf{x}_t = \tilde{\boldsymbol{\eta}}_t, \tag{57}$$

$$\tilde{\Psi}_t^{-1} = \Psi_t^{-1} + \sum_{s=1}^{p} \exp(\mathbf{x}_t^\top \hat{\mathbf{c}}_s + \hat{\mathbf{c}}_s^\top \boldsymbol{\mu}_{\mathbf{h}^{(i)}} + \tfrac{1}{2}\hat{\mathbf{c}}_s^\top \Sigma_{\mathbf{h}^{(i)}} \hat{\mathbf{c}}_s + \mathbf{d}_s) \, \hat{\mathbf{c}}_s \hat{\mathbf{c}}_s^\top, \quad \text{where } \mathbf{x}_t = \tilde{\boldsymbol{\eta}}_t. \tag{58}$$

The first term in the integrand above is given by

$$< \log(p(\mathbf{x}_t|\mathbf{x}_{t-1}) >_{q_\theta(\theta)} = -\tfrac{1}{2}(\mathbf{x}_t^\top \mathbf{x}_t - 2\mathbf{x}_t^\top (< A > \mathbf{x}_{t-1} + < B > \mathbf{u}_t))$$
$$-\tfrac{1}{2}(\mathbf{x}_{t-1}^\top < A^\top A > \mathbf{x}_{t-1} + 2\mathbf{x}_{t-1}^\top < A^\top B > \mathbf{u}_t) - \tfrac{1}{2}\mathbf{u}_t^\top < B^\top B > \mathbf{u}_t. \tag{59}$$

Therefore, the integral is given by

$$\int d\mathbf{x}_t \exp\left(< \log(p(\mathbf{x}_t|\mathbf{x}_{t-1}) >_{q_\theta(\theta)}\right) \mathcal{N}(\mathbf{x}_t|\tilde{\boldsymbol{\eta}}_t, \tilde{\Psi}_t) = \tilde{Z} \int d\mathbf{x}_t \exp\left(-\tfrac{1}{2}\mathbf{x}_t^\top(I + \tilde{\Psi}_t^{-1})\mathbf{x}_t + \mathbf{x}_t^\top(< A > \mathbf{x}_{t-1} + < B > \mathbf{u}_t + \tilde{\Psi}_t^{-1}\tilde{\boldsymbol{\eta}}_t)\right)$$

where (only showing the terms depending on $\mathbf{x}_{t-1}$)

$$\tilde{Z} = -\tfrac{1}{2}(\mathbf{x}_{t-1}^\top < A^\top A > \mathbf{x}_{t-1} + 2\mathbf{x}_{t-1}^\top < A^\top B > \mathbf{u}_t) + \cdots \tag{60}$$

After integrating out $\mathbf{x}_t$ by formulating a Gaussian distribution $\mathcal{N}(\mathbf{x}_t|\boldsymbol{\eta}_t^*, \Psi_t^*)$ where the mean and covariance are given by

$$\Psi_t^{*-1} = I + \tilde{\Psi}_t^{-1}, \tag{61}$$
$$\boldsymbol{\eta}_t^* = \Psi_t^*(< A > \mathbf{x}_{t-1} + < B > \mathbf{u}_t + \tilde{\Psi}_t^{-1}\tilde{\boldsymbol{\eta}}_t), \tag{62}$$

we obtain a quadratic function in $\mathbf{x}_{t-1}$ (combining the remainder from the integral and $\tilde{Z}$)

$$\tfrac{1}{2}(< A > \mathbf{x}_{t-1} + < B > \mathbf{u}_t + \tilde{\Psi}_t^{-1}\tilde{\boldsymbol{\eta}}_t)^\top \Psi_t^*(< A > \mathbf{x}_{t-1} + < B > \mathbf{u}_t + \tilde{\Psi}_t^{-1}\tilde{\boldsymbol{\eta}}_t) - \tfrac{1}{2}(\mathbf{x}_{t-1}^\top < A^\top A > \mathbf{x}_{t-1} + 2\mathbf{x}_{t-1}^\top < A^\top B > \mathbf{u}_t)$$
$$= -\tfrac{1}{2}(\mathbf{x}_{t-1}^\top(< A^\top A > - < A >^\top \Psi_t^* < A >)\mathbf{x}_{t-1} - 2\mathbf{x}_{t-1}^\top(< A >^\top \Psi_t^*(< B > \mathbf{u}_t + \tilde{\Psi}_t^{-1}\tilde{\boldsymbol{\eta}}_t) - < A^\top B > \mathbf{u}_t)) + \cdots. \tag{63}$$

Therefore, the backward message is approximately Gaussian with the mean and covariance given by

$$\beta(\mathbf{x}_{t-1}) \approx \mathcal{N}(\mathbf{x}_{t-1}|\boldsymbol{\eta}_{t-1}, \Psi_{t-1}), \tag{64}$$
$$\Psi_{t-1}^{-1} = < A^\top A > - < A >^\top \Psi_t^* < A >, \tag{65}$$
$$\boldsymbol{\eta}_{t-1} = \Psi_{t-1}(< A >^\top \Psi_t^*(< B > \mathbf{u}_t + \tilde{\Psi}_t^{-1}\tilde{\boldsymbol{\eta}}_t) - < A^\top B > \mathbf{u}_t). \tag{66}$$

**Computing marginals of latent variables using $\alpha$ and $\beta$**

Using the $\alpha$ and $\beta$ recursions in the forward/backward algorithm, we can compute the marginals of the latent variables.

$$p(\mathbf{x}_t|\mathbf{y}_{1:T}) = p(\mathbf{x}_t|\mathbf{y}_{1:t}, \mathbf{y}_{t+1:T}), \tag{67}$$
$$\propto p(\mathbf{y}_{t+1:T}|\mathbf{x}_t, \mathbf{y}_{1:t})p(\mathbf{x}_t|\mathbf{y}_{1:t}) = p(\mathbf{y}_{t+1:T}|\mathbf{x}_t)p(\mathbf{x}_t|\mathbf{y}_{1:t}) = \beta(\mathbf{x}_t)\alpha(\mathbf{x}_t), \tag{68}$$
$$\propto \mathcal{N}(\mathbf{x}_t|\boldsymbol{\omega}_t, \Upsilon_t) \tag{69}$$

where

$$\Upsilon_t^{-1} = \Psi_t^{-1} + \Sigma_t^{-1}, \tag{70}$$
$$\boldsymbol{\omega}_t = \Upsilon_t(\Psi_t^{-1}\boldsymbol{\eta}_t + \Sigma_t^{-1}\boldsymbol{\mu}_t). \tag{71}$$

We also need to compute pairwise marginals of latent variables, given by

$$
\begin{aligned}
p(\mathbf{x}_t, \mathbf{x}_{t+1}|\mathbf{y}_{1:T}) &= p(\mathbf{x}_t, \mathbf{x}_{t+1}|\mathbf{y}_{1:t}, \mathbf{y}_{t+1}, \mathbf{y}_{t+2:T}), \\
&\propto p(\mathbf{y}_{t+1}, \mathbf{y}_{t+2:T}|\mathbf{x}_t, \mathbf{x}_{t+1}, \mathbf{y}_{1:t})p(\mathbf{x}_{t+1}|\mathbf{x}_t, \mathbf{y}_{1:t})p(\mathbf{x}_t|\mathbf{y}_{1:t}), \\
&\propto p(\mathbf{y}_{t+1}|\mathbf{x}_{t+1})p(\mathbf{y}_{t+2:T}|\mathbf{x}_{t+1})p(\mathbf{x}_{t+1}|\mathbf{x}_t)p(\mathbf{x}_t|\mathbf{y}_{1:t}), \\
&\propto \beta(\mathbf{x}_{t+1})\exp\left(< \log(p(\mathbf{y}_{t+1}|\mathbf{x}_{t+1})p(\mathbf{x}_{t+1}|\mathbf{x}_t)) >_{q_\theta(\theta)}\right)\alpha(\mathbf{x}_t),
\end{aligned}
\tag{72}
$$

which are jointly Gaussian

$$
p\begin{pmatrix} \mathbf{x}_t \\ \mathbf{x}_{t+1} \end{pmatrix} = \mathcal{N}\left(\begin{bmatrix} \boldsymbol{\omega}_t \\ \boldsymbol{\omega}_{t+1} \end{bmatrix}, \begin{bmatrix} \Upsilon_t & \Upsilon_{t,t+1} \\ \Upsilon_{t,t+1}^T & \Upsilon_{t+1} \end{bmatrix}\right).
\tag{73}
$$

To compute the cross-covariance $\Upsilon_{t,t+1}$, we first compute the second derivatives w.r.t. $[\mathbf{x}_t\ \mathbf{x}_{t+1}]^T$:

$$
\frac{\partial^2 \log \int d\theta q_\theta(\theta)p(\mathbf{x}_t, \mathbf{x}_{t+1}|\mathbf{y}_{1:T})}{\partial[\mathbf{x}_t\ \mathbf{x}_{t+1}]^2} = -\begin{bmatrix} \Sigma_t^{*-1} & -<A>^T \\ -<A> & \Psi_{t+1}^{-1} + I + W_{t+1} \end{bmatrix},
\tag{74}
$$

where

$$
\begin{aligned}
W_{t+1} &= -\frac{\partial^2}{\partial \mathbf{x}_{t+1}\mathbf{x}_{t+1}^\top} < \log p(\mathbf{y}_{t+1}|\mathbf{x}_{t+1}) >_{q(\theta)}, \tag{75} \\
&= \sum_{s=1}^{p} \exp(\mathbf{x}_{t+1}^\top \hat{\mathbf{c}}_s + \hat{\mathbf{c}}_s^\top \boldsymbol{\mu}_{\mathbf{h}^{(i)}} + \tfrac{1}{2}\hat{\mathbf{c}}_s^\top \Sigma_{\mathbf{h}^{(i)}}\hat{\mathbf{c}}_s + \mathbf{d}_s)\,\hat{\mathbf{c}}_s\hat{\mathbf{c}}_s^\top \tag{76}
\end{aligned}
$$

evaluated at $\mathbf{x}_{t+1} = \boldsymbol{\omega}_{t+1}$. By negating and inverting the matrix, and using the *Schur* complement, we can obtain $\Upsilon_{t,t+1}$,

$$
\Upsilon_{t,t+1} = -(\Sigma_t^{*-1} - <A>^T (\Psi_{t+1}^{-1} + I + W_{t+1})^{-1} <A>)^{-1}(-<A>^T)(\Psi_{t+1}^{-1} + I + W_{t+1})^{-1}.
\tag{77}
$$

**Computing sufficient statistics of latent variables**

Using $q_\mathbf{x}(\mathbf{x}_{0:T})$, we can compute the sufficient statistics of latent variables (that are used in M step).

$$
\begin{aligned}
W_A &= \sum_{t=1}^{T} < \mathbf{x}_{t-1}\mathbf{x}_{t-1}^T > = \sum_{t=1}^{T} \Upsilon_{t-1} + \boldsymbol{\omega}_{t-1}\boldsymbol{\omega}_{t-1}^T, \qquad S_A = \sum_{t=1}^{T} < \mathbf{x}_{t-1}\mathbf{x}_t^T > = \sum_{t=1}^{T} \Upsilon_{t-1,t} + \boldsymbol{\omega}_{t-1}\boldsymbol{\omega}_t^T, \tag{78} \\
W_C &= \sum_{t=1}^{T} < \mathbf{x}_t\mathbf{x}_t^T > = \sum_{t=1}^{T} \Upsilon_t + \boldsymbol{\omega}_t\boldsymbol{\omega}_t^T, \qquad S_C = \sum_{t=1}^{T} < \mathbf{x}_t > \mathbf{y}_t^T = \sum_{t=1}^{T} \boldsymbol{\omega}_t\mathbf{y}_t^T. \tag{79}
\end{aligned}
$$

# Hyperaparameter estimation

We take the derivatives of the variational lower bound w.r.t. each hyperparameter to obtain update rules. The lower bound is simplified as below:

$$
\begin{aligned}
\log p(\mathbf{y}_{1:T}^{(1:r)}) &\geq \int d\theta\, d\mathbf{x}_{0:T}^{(1:r)}\, q(\theta, \mathbf{x}_{0:T}^{(1:r)})\, \log \frac{p(\theta, \mathbf{x}_{0:T}^{(1:r)}, \mathbf{y}_{1:T}^{(1:r)})}{q(\theta, \mathbf{x}_{0:T}^{(1:r)})}, \\
&= \int d\theta\, d\mathbf{x}_{0:T}^{(1:r)}\, q(\theta, \mathbf{x}_{0:T}^{(1:r)})\, \log p(\mathbf{x}_{0:T}^{(1:r)}, \mathbf{y}_{1:T}^{(1:r)}|\theta) - \int d\mathbf{x}_{0:T}^{(1:r)}\, q(\mathbf{x}_{0:T}^{(1:r)})\, \log q(\mathbf{x}_{0:T}^{(1:r)}) + \int d\theta\, d\mathbf{x}_{0:T}^{(1:r)}\, q(\theta, \mathbf{x}_{0:T}^{(1:r)})\, \log \frac{p(\theta)}{q(\theta)}, \\
&= \sum_{i=1}^{r} \log Z'_{(i)} + \int d\theta\, d\mathbf{x}_{0:T}^{(1:r)}\, q(\theta, \mathbf{x}_{0:T}^{(1:r)})\, \log \frac{p(\theta)}{q(\theta)}, \tag{80}
\end{aligned}
$$

where the last line follows from the equality

$$-\int d\mathbf{x}_{0:T}^{(1:r)} q_\mathbf{x}(\mathbf{x}_{0:T}^{(1:r)}) \log q_\mathbf{x}(\mathbf{x}_{0:T}^{(1:r)}) = -\int d\mathbf{x}_{0:T}^{(1:r)} q_\mathbf{x}(\mathbf{x}_{0:T}^{(1:r)})\mathbb{E}_{q_\theta(\theta)} \log p(\mathbf{x}_{0:T}^{(1:r)}, \mathbf{y}_{1:T}^{(1:r)}|\theta) + \sum_{i=1}^{r} \log Z'_{(i)}. \tag{81}$$

So, we need to consider the second term in RHS of the lower bound for hyperparameter update (the integration w.r.t. $\mathbf{x}$ is omitted, since the integrand is independent of $\mathbf{x}$)

$$\int d\mathbf{a}\, d\mathbf{b}\, d\mathbf{h}^{(1:r)} q(\mathbf{a}, \mathbf{b}) q(\mathbf{h}^{(1:r)}) \log \frac{p(\mathbf{a}, \mathbf{b}, \mathbf{h}^{(1:r)})}{q(\mathbf{a}, \mathbf{b}, \mathbf{h}^{(1:r)})} = -KL(\mathbf{a}, \mathbf{b}) - KL(\mathbf{h}^{(1:r)}), \tag{82}$$

where the first term on RHS is given by

$$KL(\mathbf{a}, \mathbf{b}) = \int d\mathbf{a}\, d\mathbf{b} q(\mathbf{a}, \mathbf{b}) \log \frac{q(\mathbf{a}, \mathbf{b})}{p(\mathbf{a}, \mathbf{b})}, \tag{83}$$

$$= \int d\mathbf{a}\, d\mathbf{b} \mathcal{N}(\boldsymbol{\mu}_{\mathbf{a},\mathbf{b}}, \Lambda_{\mathbf{a},\mathbf{b}}) \log \frac{\mathcal{N}(\boldsymbol{\mu}_{\mathbf{a},\mathbf{b}}, \Lambda_{\mathbf{a},\mathbf{b}})}{\mathcal{N}(0, \tilde{\Lambda}_{\mathbf{a},\mathbf{b}})}, \tag{84}$$

$$= -\tfrac{1}{2} \log |\tilde{\Lambda}_{\mathbf{a},\mathbf{b}}^{-1} \Lambda_{\mathbf{a},\mathbf{b}}| + \tfrac{1}{2}\text{Tr}[\tilde{\Lambda}_{\mathbf{a},\mathbf{b}}^{-1}(\Lambda_{\mathbf{a},\mathbf{b}} + \boldsymbol{\mu}_{\mathbf{a},\mathbf{b}}\boldsymbol{\mu}_{\mathbf{a},\mathbf{b}}^{\top})] \tag{85}$$

where the prior covariance on $(\mathbf{a}, \mathbf{b})$ is denoted by $\tilde{\Lambda}_{\mathbf{a},\mathbf{b}} = [\alpha^{-1}I\ 0; 0\ \beta^{-1}I]$ and the posterior mean and covariance on $(\mathbf{a}, \mathbf{b})$ are denoted by $\boldsymbol{\mu}_{\mathbf{a},\mathbf{b}} = [\boldsymbol{\mu}_\mathbf{a}; \boldsymbol{\mu}_\mathbf{b}]$ and $\Lambda_{\mathbf{a},\mathbf{b}} = [\Sigma_\mathbf{a}\ \Sigma_{\mathbf{a},\mathbf{b}}; \Sigma_{\mathbf{a},\mathbf{b}}^{\top}\ \Sigma_\mathbf{b}]$, respectively. We minimise the KL divergence for updating $\alpha, \beta$.

The second term on RHS is given by

$$KL(\mathbf{h}^{(1:r)}) = \int d\mathbf{h}^{(1:r)} q_\mathbf{h}(\mathbf{h}^{(1:r)}) \log \frac{q_\mathbf{h}(\mathbf{h}^{(1:r)})}{p(\mathbf{h}^{(1:r)}|\sigma^2, \tau^2)}, \tag{86}$$

$$= \int d\mathbf{h}^{(1:r)} \mathcal{N}(\mathbf{h}^{(1:r)}|\boldsymbol{\mu}_\mathbf{h}, \Sigma_\mathbf{h}) \log \frac{\mathcal{N}(\mathbf{h}^{(1:r)}|\boldsymbol{\mu}_\mathbf{h}, \Sigma_\mathbf{h})}{p(\mathbf{h}^{(1:r)}|\mathbf{m}_\mathbf{h}, K)}, \tag{87}$$

$$= -\tfrac{1}{2} \log |K^{-1}\Sigma_\mathbf{h}| + \tfrac{1}{2}\text{Tr}\left[K^{-1}(\Sigma_\mathbf{h} + (\boldsymbol{\mu}_\mathbf{h} - \mathbf{m}_\mathbf{h})(\boldsymbol{\mu}_\mathbf{h} - \mathbf{m}_\mathbf{h})^{\top})\right] + const. \tag{88}$$

The first derivative w.r.t. kernel parameters (denoted by $\boldsymbol{\alpha} = \{\sigma^2, \tau^2\}$) is given by

$$\frac{\partial}{\partial \boldsymbol{\alpha}} KL(\mathbf{h}^{(1:r)}) = \tfrac{1}{2}\text{Tr}\left(K^{-1}\frac{\partial K}{\partial \boldsymbol{\alpha}}\right) - \tfrac{1}{2}\text{Tr}\left(K^{-1}\frac{\partial K}{\partial \boldsymbol{\alpha}}K^{-1}(\Sigma_\mathbf{h} + (\boldsymbol{\mu}_\mathbf{h} - \mathbf{m}_\mathbf{h})(\boldsymbol{\mu}_\mathbf{h} - \mathbf{m}_\mathbf{h})^{\top})\right), \tag{89}$$

$$= \tfrac{1}{2}\text{Tr}\left(K^{-1}\frac{\partial K}{\partial \boldsymbol{\alpha}}(I - K^{-1}(\Sigma_\mathbf{h} + (\boldsymbol{\mu}_\mathbf{h} - \mathbf{m}_\mathbf{h})(\boldsymbol{\mu}_\mathbf{h} - \mathbf{m}_\mathbf{h})^{\top}))\right), \tag{90}$$

where the first derivative of $K(i, j)$ w.r.t. $\boldsymbol{\alpha}$ is given by

$$\frac{\partial}{\partial \tau^2} K(i, j) = \frac{1}{2\tau^4}(i - j)^2(\sigma^2 + \epsilon\delta_{ij})\exp\left(-\frac{1}{2\tau^2}(i - j)^2\right) I_{k^2} = \frac{1}{2\tau^4}(i - j)^2 K(i, j), \tag{91}$$

$$\frac{\partial}{\partial \sigma^2} K(i, j) = \exp\left(-\frac{1}{2\tau^2}(i - j)^2\right) I_{k^2}. \tag{92}$$

We update $\boldsymbol{\alpha}$ numerically using the derivative expression above.

# Model II: nonstationarity in neural dynamics

## Basic setup

**Likelihood**: $\mathbf{y}_t \in \mathbb{R}^p$, $\mathbf{x}_t \in \mathbb{R}^k$, $C \in \mathbb{R}^{p \times k}$

$$p(\mathbf{y}_t|\mathbf{x}_t, C, \mathbf{d}) = \text{Poiss}(\mathbf{y}_t|\exp(C\mathbf{x}_t + \mathbf{d})).$$

**Latent dynamics**: $A \in \mathbb{R}^{k \times k}$

$$p(\mathbf{x}_t|\mathbf{x}_{t-1}, A) = \mathcal{N}(\mathbf{x}_t|A\mathbf{x}_{t-1}, I).$$

Parameters in this model: $\Theta = \{A, C\}$. For simplicity, we will fix $\mathbf{d}$ to its maximum likelihood estimate. Vectorized notations: $\mathbf{a} = \text{vec}(A^\top) \in \mathbb{R}^{k^2}$ and $\mathbf{c} = \text{vec}(C^\top) \in \mathbb{R}^{pk}$.

**Priors**:

$$p(\mathbf{c}|\gamma) = \mathcal{N}(\mathbf{c}|\mathbf{0}, \gamma^{-1}\mathbf{I}) \tag{93}$$

Assuming $\mathbf{a}$ to be temporally evolving across recordings where the recording index is $i = \{1, \cdots, r\}$:

$$\mathbf{a}^{(i)} \quad \sim \quad \mathcal{GP}(\bar{\mathbf{a}}, K(i,j)) \tag{94}$$

where we denote the (vector) mean and (matrix) covariance functions by $\bar{\mathbf{a}}$ and $K(i,j)$, respectively, where the $(i,j)$th block of the covariance matrix is given by

$$K(i,j) \quad = \quad (\sigma^2 + \epsilon\delta_{i,j})\exp\left(-\frac{1}{2\tau^2}(i-j)^2\right)I_{k^2}. \tag{95}$$

The hyperparameters in total are $\Phi = \{\bar{\mathbf{a}}, \sigma^2, \tau^2, \gamma\}$.

## Variational lower bound

The marginal likelihood of the observations is lower bounded by

$$\log p(\mathbf{y}_{1:T}^{(1:r)}) \quad \geq \quad \int d\theta \, d\mathbf{x}_{0:T}^{(1:r)} \, q(\theta, \mathbf{x}_{0:T}^{(1:r)}) \, \log \frac{p(\theta, \mathbf{x}_{0:T}^{(1:r)}, \mathbf{y}_{1:T}^{(1:r)})}{q(\theta, \mathbf{x}_{0:T}^{(1:r)})}, \tag{96}$$

where the approximate posterior factories

$$q(\theta, \mathbf{x}_{0:T}^{(1:r)}) = q_\theta(\theta) \prod_{i=1}^{r} q_\mathbf{x}(\mathbf{x}_{0:T}^{(i)}), \tag{97}$$

and we assume $q_\theta(\theta) = q_\mathbf{a}(\mathbf{a}^{(1:r)})q_\mathbf{c}(\mathbf{c})$.

# Bayesian Laplace propagation

## Posterior over parameters

We compute $q_\theta(\theta)$ by integrating out latent variables from the total log joint distribution:

$$
\begin{aligned}
\log q_\theta(\theta) &= \mathbb{E}_{q_\mathbf{x}(\mathbf{x}_{0:T}^{(1:r)})}\left[\log p(\mathbf{x}_{0:T}^{(1:r)}, \mathbf{y}_{1:T}^{(1:r)}, \theta)\right] + const, \\
&= \mathbb{E}_{q_\mathbf{x}(\mathbf{x}_{0:T}^{(1:r)})}\left[\log p(\mathbf{y}_{1:T}^{(1:r)}|\mathbf{x}_{0:T}^{(1:r)}, \theta) + \log p(\mathbf{x}_{0:T}^{(1:r)}|\theta) + \log p(\theta)\right] + const, \\
&= \sum_{i=1}^{r}\left[\mathbb{E}_{q_\mathbf{x}(\mathbf{x}_{0:T}^{(i)})}(\sum_{t=1}^{T}(\log p(\mathbf{y}_t^{(i)}|\mathbf{x}_t^{(i)}, \mathbf{c}) + \log p(\mathbf{x}_t^{(i)}|\mathbf{x}_{t-1}^{(i)}, \mathbf{a}^{(i)})))\right] + \log p(\mathbf{a}^{(1:r)}|\bar{\mathbf{a}}^{(1:r)}, K) + \log p(\mathbf{c}|\gamma) + const,
\end{aligned}
\tag{98}
$$

where $\bar{\mathbf{a}}^{(1:r)}$ is a vector of $r$ repeating $\bar{\mathbf{a}}$.

### 1. approximate posterior over $\mathbf{a}^{(1:r)}$

$$
\begin{aligned}
\log q_\mathbf{a}(\mathbf{a}^{(1:r)}) &= \sum_{i=1}^{r}\left[\mathbb{E}_{q_\mathbf{x}(\mathbf{x}_{0:T}^{(i)})}\sum_{t=1}^{T}\log p(\mathbf{x}_t^{(i)}|\mathbf{x}_{t-1}^{(i)}, \mathbf{a}^{(i)})\right] + \log p(\mathbf{a}^{(1:r)}|\bar{\mathbf{a}}^{(1:r)}, K) + const, \tag{99} \\
&= -\tfrac{1}{2}(\mathbf{a}^{(1:r)\top}H\mathbf{a}^{(1:r)} - 2\mathbf{a}^{(1:r)\top}\mathbf{s}) - \tfrac{1}{2}(\mathbf{a}^{(1:r)} - \bar{\mathbf{a}}^{(1:r)})^T K^{-1}(\mathbf{a}^{(1:r)} - \bar{\mathbf{a}}^{(1:r)}) \tag{100}
\end{aligned}
$$

where the matrix $H$ and the vector $\mathbf{s}$ are given by

$$
H_\mathbf{a} = \begin{pmatrix} W_{A^{(1)}}^{bd}, & 0, & 0, & \dots & 0 \\ 0, & W_{A^{(2)}}^{bd}, & 0, & \dots & 0 \\ \vdots & \vdots & \vdots & \vdots & \vdots \\ 0, & \dots & \dots, & 0, & W_{A^{(r)}}^{bd} \end{pmatrix}, \quad \mathbf{s} = \begin{pmatrix} \text{vec}(S_{A^{(1)}}) \\ \vdots \\ \text{vec}(S_{A^{(r)}}) \end{pmatrix}
\tag{101}
$$

and $W_{A^{(i)}}^{bd} = I_k \otimes W_{A^{(i)}}$, where $W_{A^{(i)}} = \sum_{t=1}^{T} <\mathbf{x}_{t-1}^{(i)}\mathbf{x}_{t-1}^{(i)\top}>$, and $S_{A^{(i)}} = \sum_{t=1}^{T} <\mathbf{x}_{t-1}^{(i)}\mathbf{x}_t^{(i)\top}>$.

Therefore, the approximate posterior over $\mathbf{a}^{(1:r)}$ is given by

$$
\begin{aligned}
q(\mathbf{a}^{(1:r)}) &= \mathcal{N}(\boldsymbol{\mu}_\mathbf{a}, \Sigma_\mathbf{a}), \tag{102} \\
\Sigma_\mathbf{a}^{-1} &= K^{-1} + H_\mathbf{a}, \tag{103} \\
\boldsymbol{\mu}_\mathbf{a} &= \Sigma_\mathbf{a}(K^{-1}\bar{\mathbf{a}}^{(1:r)} + \mathbf{s}). \tag{104}
\end{aligned}
$$

So, $< A^{(i)} >= \left[\text{reshape}(\boldsymbol{\mu}_\mathbf{a}((i-1)k^2 + 1 : ik^2), k, k)\right]^\top$ and $\Sigma_{A^{(i)}}$ is the first $k \times k$ matrix of $\Sigma_\mathbf{a}((i-1)k^2+1 : ik^2, (i-1)k^2+1 : ik^2)$. In addition to the mean and covariance of $A^{(i)}$, we also need the following quantity in VBE step:

$$
< A^{(i)\top}A^{(i)} > = < A^{(i)} >^\top < A^{(i)} > +k\Sigma_{A^{(i)}}. \tag{105}
$$

## 2. Computing $q_{\mathbf{c}}(\mathbf{c})$

Similarly, we write down all the terms in $\log p(\mathbf{x}_{0:T}, \mathbf{y}_{1:T}, \theta)$ that depend on $\mathbf{c}$:

$$
\begin{aligned}
\log q_{\mathbf{c}}(\mathbf{c}) &= \sum_{i=1}^{r} \mathbb{E}_{q_{\mathbf{x}}(\mathbf{x}_{1:T}^{(i)})}\left[\sum_{t=1}^{T} \log p(\mathbf{y}_t^{(i)}|\mathbf{x}_t^{(i)}, \mathbf{c})\right] + \log p(\mathbf{c}|\gamma) + const, \\
&= \sum_{i=1}^{r} \mathbb{E}_{q_{\mathbf{x}}(\mathbf{x}_{1:T}^{(i)})}\left[\sum_{t=1}^{T}(\mathbf{y}_t^{(i)\top}(C\mathbf{x}_t^{(i)} + \mathbf{d}) - \mathbf{1}^{\top}\exp{(C\mathbf{x}_t^{(i)} + \mathbf{d})})\right] - \frac{1}{2}\gamma\mathbf{c}^T\mathbf{c} + const, \\
&= \sum_{i=1}^{r}\left[\mathbf{c}^{\top}\mathrm{vec}(S_{C^{(i)}}) - \sum_{s=1}^{p} \mathbb{E}_{q_{\mathbf{x}}(\mathbf{x}_{1:T}^{(i)})}\{\sum_{t=1}^{T}\exp(\mathbf{c}_s^{\top}\mathbf{x}_t^{(i)} + \mathbf{d}_s)\}\right] - \frac{1}{2}\gamma\mathbf{c}^{\top}\mathbf{c} + const, \quad (106)
\end{aligned}
$$

where each row of $C$ is denoted by $\mathbf{c}_s$ and the sufficient statistic is denoted by

$$
S_{C^{(i)}} = \sum_{t=1}^{T} <\mathbf{x}_t^{(i)}> \mathbf{y}_t^{(i)\top} \quad (107)
$$

Assuming the approximate posterior over latent variables is multivariate Gaussian with marginals $q(\mathbf{x}_t) = \mathcal{N}(\mathbf{x}_t|\boldsymbol{\omega}_t, \Upsilon_t)$, the expectation of the exponential term in eq. 106 is given by

$$
\begin{aligned}
\mathbb{E}_{q_{\mathbf{x}}(\mathbf{x}_{1:T}^{(i)})}\{\sum_{t=1}^{T}\exp(\mathbf{c}_s^{\top}\mathbf{x}_t^{(i)})\} &= \int d\mathbf{x}_{1:T}^{(i)} q_{\mathbf{x}}(\mathbf{x}_{1:T}^{(i)})\exp(\mathbf{c}_s^{\top}\mathbf{x}_1^{(i)} + \cdots + \mathbf{c}_s^{\top}\mathbf{x}_T^{(i)}), \\
&= \sum_{t=1}^{T}\exp(\mathbf{c}_s^{\top}\boldsymbol{\omega}_t^{(i)} + \tfrac{1}{2}\mathbf{c}_s^{\top}\Upsilon_t^{(i)}\mathbf{c}_s). \quad (108)
\end{aligned}
$$

Therefore, the log joint distribution is given by

$$
\log q_{\mathbf{c}}(\mathbf{c}) = \sum_{i=1}^{r}\left[\mathbf{c}^{\top}\mathrm{vec}(S_{C^{(i)}}) - \sum_{s=1}^{p}\sum_{t=1}^{T}\exp(\mathbf{c}_s^{\top}\boldsymbol{\omega}_t^{(i)} + \tfrac{1}{2}\mathbf{c}_s^{\top}\Upsilon_t^{(i)}\mathbf{c}_s + \mathbf{d}_s)\right] - \frac{1}{2}\gamma\mathbf{c}^{\top}\mathbf{c} + const. \quad (109)
$$

We approximate $q_C(C)$ to a Gaussian distribution from the first/second derivatives of eq. 109 w.r.t. $\mathbf{c}_s$,

$$
q_C(C) = \prod_{s=1}^{p}\mathcal{N}(\mathbf{c}_s|\boldsymbol{\mu}_{\mathbf{c}_s}, \Sigma_{\mathbf{c}_s}) \quad (110)
$$

$$
\boldsymbol{\mu}_{\mathbf{c}_s} = \frac{1}{\gamma}\sum_{i=1}^{r}\left[S_{C^{(i)}}\mathbf{e}_s - \sum_{t=1}^{T}[\boldsymbol{\omega}_t^{(i)} + \Upsilon_t^{(i)}\mathbf{c}_s]\exp(\mathbf{c}_s^T\boldsymbol{\omega}_t^{(i)} + \tfrac{1}{2}\mathbf{c}_s^T\Upsilon_t^{(i)}\mathbf{c}_s + \mathbf{d}_s)\right], \quad \text{where } \mathbf{c}_s = \boldsymbol{\mu}_{\mathbf{c}_s}, \quad (111)
$$

$$
\Sigma_{\mathbf{c}_s}^{-1} = \gamma I + \sum_{i=1}^{r}\sum_{t=1}^{T}(\Upsilon_t^{(i)} + (\boldsymbol{\omega}_t^{(i)} + \Upsilon_t^{(i)}\mathbf{c}_s)(\boldsymbol{\omega}_t^{(i)} + \Upsilon_t^{(i)}\mathbf{c}_s)^{\top})\exp(\mathbf{c}_s^T\boldsymbol{\omega}_t^{(i)} + \tfrac{1}{2}\mathbf{c}_s^T\Upsilon_t^{(i)}\mathbf{c}_s + \mathbf{d}_s), \quad \text{where } \mathbf{c}_s = \boldsymbol{\mu}_{\mathbf{c}_s}. \quad (112)
$$

## 3. ML estimate of d

The ML estimate of $\mathbf{d}$ given the mean of $C$ (denoted by $\hat{C}$) is closed form:

$$
\begin{aligned}
\hat{\mathbf{d}} &= \arg\max_{\mathbf{d}} \sum_{i=1}^{r}\left[\hat{C}S_{C^{(i)}} + \sum_{t=1}^{T}\left(\mathbf{y}_t^{(i)\top}\mathbf{d} - \mathbf{1}^\top \exp(\hat{C}\boldsymbol{\omega}_t^{(i)} + \tfrac{1}{2}\mathrm{diag}(\hat{C}\Upsilon_t^{(i)}\hat{C}^\top) + \mathbf{d}))\right], \\
&= \log(\sum_{i=1}^{r}\sum_{t=1}^{T}\mathbf{y}_t^{(i)}) - \log(\sum_{i=1}^{r}\sum_{t=1}^{T}\exp(\hat{C}\boldsymbol{\omega}_t^{(i)} + \tfrac{1}{2}\mathrm{diag}(\hat{C}\Upsilon_t^{(i)}\hat{C}^\top))).
\end{aligned} \tag{113}
$$

## Posterior over latent variables

We compute $q_{\mathbf{x}}(\mathbf{x}_{0:qT})$ by

$$
\begin{aligned}
\sum_{i=1}^{r}\log q_{\mathbf{x}}(\mathbf{x}_{0:T}^{(i)}) &= \sum_{i=1}^{r}\mathbb{E}_{q_\theta(\theta)}\log p(\theta, \mathbf{x}_{0:T}^{(i)}, \mathbf{y}_{1:T}^{(i)}) + const, \\
&= \sum_{i=1}^{r}\mathbb{E}_{q_\theta(\theta)}\log p(\mathbf{x}_{0:T}^{(i)}, \mathbf{y}_{1:T}^{(i)}|\theta) - \sum_{i=1}^{r}\log Z'_{(i)},
\end{aligned} \tag{114}
$$

where the normalization constant is given by

$$
Z'_{(i)} = \int d\mathbf{x}_{0:T}^{(i)}\exp\left(\mathbb{E}_{q_\theta(\theta)}\log p(\mathbf{x}_{0:T}^{(i)}, \mathbf{y}_{1:T}^{(i)}|\theta)\right). \tag{115}
$$

The complete-data log likelihood in the $i$th recording is written as

$$
\log p(\mathbf{x}_{0:T}^{(i)}, \mathbf{y}_{1:T}^{(i)}|\theta) = \sum_{t=1}^{T}\{\log p(\mathbf{y}_t^{(i)}|\mathbf{x}_t^{(i)}, C, \mathbf{d}) + \log p(\mathbf{x}_t^{(i)}|\mathbf{x}_{t-1}^{(i)}, A^{(i)})\}, \tag{116}
$$

which tells us that the log posterior over latent variables is quadratic in each $\mathbf{x}_t$. This enables us to use the sequential update of the posterior over latent variables. We will also use the following sequential forward/backward algorithm for each recording in parallel. In the following, the recording index $i$ is removed for notational cleanness.

### Forward filtering

We denote the posterior over the latent variables at each time $t$ by

$$
\alpha(\mathbf{x}_t) \propto \int d\mathbf{x}_{t-1}\alpha(\mathbf{x}_{t-1})\exp\left[<\log(p(\mathbf{x}_t|\mathbf{x}_{t-1})p(\mathbf{y}_t|\mathbf{x}_t))>_{q_\theta(\theta)}\right], \tag{117}
$$

$$
\propto \exp(<\log p(\mathbf{y}_t|\mathbf{x}_t))>_{q_\theta(\theta)})\left\{\int d\mathbf{x}_{t-1}\alpha(\mathbf{x}_{t-1})\exp\left(<\log(p(\mathbf{x}_t|\mathbf{x}_{t-1})>_{q(\theta)})\right)\right\}. \tag{118}
$$

Assuming $\alpha(\mathbf{x}_{t-1}) = \mathcal{N}(\mathbf{x}_{t-1}|\boldsymbol{\mu}_{t-1}, \Sigma_{t-1})$, the integral is analytically tractable since the second part in the integrand is also quadratic in $\mathbf{x}_{t-1}$:

$$
\exp[-\tfrac{1}{2}(\mathbf{x}_{t-1}^\top < A^\top A > \mathbf{x}_{t-1} - 2\mathbf{x}_{t-1}^\top < A >^\top \mathbf{x}_t + \mathbf{x}_t^\top \mathbf{x}_t)].
$$

The integrand is summarised as

$$
\begin{aligned}
\alpha(\mathbf{x}_{t-1})\exp\left(<\log(p(\mathbf{x}_t|\mathbf{x}_{t-1})>_{q(\theta)}\right) &= Z\mathcal{N}(\mathbf{x}_{t-1}|\boldsymbol{\mu}_{t-1}^*, \Sigma_{t-1}^*), \tag{119} \\
\Sigma_{t-1}^{*-1} &= \Sigma_{t-1}^{-1} + < A^\top A >, \tag{120} \\
\boldsymbol{\mu}_{t-1}^* &= \Sigma_{t-1}^*(\Sigma_{t-1}^{-1}\mu_{t-1} + < A >^\top \mathbf{x}_t), \tag{121}
\end{aligned}
$$

and the remaining term $Z$ is given by:

$$Z = \exp[\tfrac{1}{2}{\boldsymbol{\mu}_{t-1}^*}^\top \Sigma_{t-1}^{*-1} \boldsymbol{\mu}_{t-1}^*], \tag{122}$$

where

$$
\begin{aligned}
\tfrac{1}{2}{\boldsymbol{\mu}_{t-1}^*}^\top \Sigma_{t-1}^{*-1} \boldsymbol{\mu}_{t-1}^* &= \tfrac{1}{2}(\Sigma_{t-1}^{-1}\boldsymbol{\mu}_{t-1} + <A>^\top \mathbf{x}_t)^\top \Sigma_{t-1}^*(\Sigma_{t-1}^{-1}\boldsymbol{\mu}_{t-1} + <A>^\top \mathbf{x}_t), \\
&= \tfrac{1}{2}({\mathbf{x}_t}^\top <A> \Sigma_{t-1}^* <A>^\top \mathbf{x}_t + 2{\mathbf{x}_t}^\top <A> \Sigma_{t-1}^* \Sigma_{t-1}^{-1}\boldsymbol{\mu}_{t-1}) + \cdots .
\end{aligned}
$$

Therefore, $Z$ is proportional to a Gaussian in $\mathbf{x}_t$ :

$$Z \propto \mathcal{N}(\mathbf{x}_t | \tilde{\boldsymbol{\mu}}_t, \tilde{\Sigma}_t), \tag{123}$$
$$\tilde{\Sigma}_t^{-1} = I - <A> \Sigma_{t-1}^* <A>^\top, \tag{124}$$
$$\tilde{\boldsymbol{\mu}}_t = \tilde{\Sigma}_t <A> \Sigma_{t-1}^* \Sigma_{t-1}^{-1} \boldsymbol{\mu}_{t-1}, \tag{125}$$

We approximate the forward message as a Gaussian in $\mathbf{x}_t$ using the first and second derivatives w.r.t. $\mathbf{x}_t$

$$\alpha(\mathbf{x}_t) \propto \exp(< \log p(\mathbf{y}_t|\mathbf{x}_t) >_{q_\theta(\theta)}) \mathcal{N}(\mathbf{x}_t | \tilde{\boldsymbol{\mu}}_t, \tilde{\Sigma}_t). \tag{126}$$

The forward message at time $t$ is approximately

$$\alpha(\mathbf{x}_t) \approx \mathcal{N}(\mathbf{x}_t | \boldsymbol{\mu}_t, \Sigma_t), \tag{127}$$
$$\boldsymbol{\mu}_t = \tilde{\boldsymbol{\mu}}_t + \tilde{\Sigma}_t \sum_{s=1}^{p} \left[ (\mathbf{y}_t^T \mathbf{e}_s)\boldsymbol{\mu}_{\mathbf{c}_s} - (\boldsymbol{\mu}_{\mathbf{c}_s} + \Sigma_{\mathbf{c}_s}\mathbf{x}_t)e^{\mathbf{x}_t^T \boldsymbol{\mu}_{\mathbf{c}_s} + \frac{1}{2}\mathbf{x}_t^T \Sigma_{\mathbf{c}_s}\mathbf{x}_t + \mathbf{d}_s} \right], \quad \text{where } \mathbf{x}_t = \boldsymbol{\mu}_t, \tag{128}$$

$$\Sigma_t^{-1} = \tilde{\Sigma}_t^{-1} + \sum_{s=1}^{p} \left[ \Sigma_{\mathbf{c}_s} + (\boldsymbol{\mu}_{\mathbf{c}_s} + \Sigma_{\mathbf{c}_s}\mathbf{x}_t)(\boldsymbol{\mu}_{\mathbf{c}_s} + \Sigma_{\mathbf{c}_s}\mathbf{x}_t)^T \right] e^{\mathbf{x}_t^T \boldsymbol{\mu}_{\mathbf{c}_s} + \frac{1}{2}\mathbf{x}_t^T \Sigma_{\mathbf{c}_s}\mathbf{x}_t + \mathbf{d}_s}, \quad \text{where } \mathbf{x}_t = \boldsymbol{\mu}_t. \tag{129}$$

**Backward smoothing**

We denote the backward message at each time $t$ by

$$\beta(\mathbf{x}_t) = p(\mathbf{y}_{t+1:T}|\mathbf{x}_t) = \mathcal{N}(\mathbf{x}_t | \boldsymbol{\eta}_t, \Psi_t). \tag{130}$$

We can obtain the recursion rules by considering $\beta(\mathbf{x}_{t-1})$

$$
\begin{aligned}
\beta(\mathbf{x}_{t-1}) &= \int d\mathbf{x}_t \beta(\mathbf{x}_t) \exp\left( < \log(p(\mathbf{x}_t|\mathbf{x}_{t-1})p(\mathbf{y}_t|\mathbf{x}_t)) >_{q_\theta(\theta)} \right), \\
&= \int d\mathbf{x}_t \exp\left( < \log(p(\mathbf{x}_t|\mathbf{x}_{t-1}) >_{q_\theta(\theta)} \right) \left[ \beta(\mathbf{x}_t) \exp\left( < \log p(\mathbf{y}_t|\mathbf{x}_t) >_{q_\theta(\theta)} \right) \right], \\
&= \int d\mathbf{x}_t \exp\left( < \log(p(\mathbf{x}_t|\mathbf{x}_{t-1}) >_{q_\theta(\theta)} \right) \mathcal{N}(\mathbf{x}_t | \tilde{\boldsymbol{\eta}}_t, \tilde{\Psi}_t),
\end{aligned} \tag{131}
$$

assuming $\beta(\mathbf{x}_T) = 1$. The Gaussian $p(\mathbf{x}_t) = \mathcal{N}(\mathbf{x}_t | \tilde{\boldsymbol{\eta}}_t, \tilde{\Psi}_t)$ is obtained by computing the first and second derivatives w.r.t. $\mathbf{x}_t$,

$$\tilde{\boldsymbol{\eta}}_t = \boldsymbol{\eta}_t + \Psi_t \sum_{s=1}^{p} \left[ (\mathbf{y}_t^T \mathbf{e}_s)\boldsymbol{\mu}_{\mathbf{c}_s} - (\boldsymbol{\mu}_{\mathbf{c}_s} + \Sigma_{\mathbf{c}_s}\mathbf{x}_t)e^{\mathbf{x}_t^T \boldsymbol{\mu}_{\mathbf{c}_s} + \frac{1}{2}\mathbf{x}_t^T \Sigma_{\mathbf{c}_s}\mathbf{x}_t + \mathbf{d}_s} \right], \quad \text{where } \mathbf{x}_t = \tilde{\boldsymbol{\eta}}_t, \tag{132}$$

$$\tilde{\Psi}_t^{-1} = \Psi_t^{-1} + \sum_{s=1}^{p} \left[ \Sigma_{\mathbf{c}_s} + (\boldsymbol{\mu}_{\mathbf{c}_s} + \Sigma_{\mathbf{c}_s}\mathbf{x}_t)(\boldsymbol{\mu}_{\mathbf{c}_s} + \Sigma_{\mathbf{c}_s}\mathbf{x}_t)^T \right] e^{\mathbf{x}_t^T \boldsymbol{\mu}_{\mathbf{c}_s} + \frac{1}{2}\mathbf{x}_t^T \Sigma_{\mathbf{c}_s}\mathbf{x}_t + \mathbf{d}_s}, \quad \text{where } \mathbf{x}_t = \tilde{\boldsymbol{\eta}}_t. \tag{133}$$

The first term in the integrand in eq. 131 is given by

$$< \log(p(\mathbf{x}_t|\mathbf{x}_{t-1})) >_{q_\theta(\theta)} \quad = \quad -\tfrac{1}{2}(\mathbf{x}_t^T \mathbf{x}_t - 2\mathbf{x}_t^T < A > \mathbf{x}_{t-1}) - \tfrac{1}{2}\mathbf{x}_{t-1}^T < A^T A > \mathbf{x}_{t-1}. \tag{134}$$

Therefore, the integral is given by

$$\int d\mathbf{x}_t \exp\left(< \log(p(\mathbf{x}_t|\mathbf{x}_{t-1})) >_{q_\theta(\theta)}\right) \mathcal{N}(\mathbf{x}_t|\tilde{\boldsymbol{\eta}}_t, \tilde{\Psi}_t) \quad = \quad \tilde{Z} \int d\mathbf{x}_t \exp\left(-\tfrac{1}{2}\mathbf{x}_t^T(I + \tilde{\Psi}_t^{-1})\mathbf{x}_t + \mathbf{x}_t^T(< A > \mathbf{x}_{t-1} + \tilde{\Psi}_t^{-1}\tilde{\boldsymbol{\eta}}_t)\right)$$

where (only showing the terms depending on $\mathbf{x}_{t-1}$)

$$\tilde{Z} \quad = \quad -\tfrac{1}{2}\mathbf{x}_{t-1}^T < A^T A > \mathbf{x}_{t-1} + \cdots \tag{135}$$

After integrating out $\mathbf{x}_t$ by formulating a Gaussian distribution $\mathcal{N}(\mathbf{x}_t|\boldsymbol{\eta}_t^*, \Psi_t^*)$ where the mean and covariance are given by

$$\Psi_t^{*-1} \quad = \quad I + \tilde{\Psi}_t^{-1}, \tag{136}$$
$$\boldsymbol{\eta}_t^* \quad = \quad \Psi_t^*(< A > \mathbf{x}_{t-1} + \tilde{\Psi}_t^{-1}\tilde{\boldsymbol{\eta}}_t), \tag{137}$$

we obtain a quadratic function in $\mathbf{x}_{t-1}$ (combining the remainder from the integral and $\tilde{Z}$)

$$\tfrac{1}{2}(< A > \mathbf{x}_{t-1} + \tilde{\Psi}_t^{-1}\tilde{\boldsymbol{\eta}}_t)^T \Psi_t^*(< A > \mathbf{x}_{t-1} + \tilde{\Psi}_t^{-1}\tilde{\boldsymbol{\eta}}_t) - \tfrac{1}{2}\mathbf{x}_{t-1}^T < A^T A > \mathbf{x}_{t-1}$$
$$= \quad -\tfrac{1}{2}(\mathbf{x}_{t-1}^T(< A^T A > - < A >^T \Psi_t^* < A >)\mathbf{x}_{t-1} - 2\mathbf{x}_{t-1}^T < A >^T \Psi_t^* \tilde{\Psi}_t^{-1}\tilde{\boldsymbol{\eta}}_t + \cdots . \tag{138}$$

Therefore, the backward message is approximately Gaussian with the mean and covariance given by

$$\beta(\mathbf{x}_{t-1}) \quad \approx \quad \mathcal{N}(\mathbf{x}_{t-1}|\boldsymbol{\eta}_{t-1}, \Psi_{t-1}), \tag{139}$$
$$\Psi_{t-1}^{-1} \quad = \quad < A^T A > - < A >^T \Psi_t^* < A >, \tag{140}$$
$$\boldsymbol{\eta}_{t-1} \quad = \quad \Psi_{t-1} < A >^T \Psi_t^* \tilde{\Psi}_t^{-1}\tilde{\boldsymbol{\eta}}_t \quad = \quad \Psi_{t-1} < A >^T (I + \tilde{\Psi}_t)^{-1}\tilde{\boldsymbol{\eta}}_t. \tag{141}$$

## Computing marginals of latent variables using $\alpha$ and $\beta$

Using the $\alpha$ and $\beta$ recursions in the forward/backward algorithm, we can compute the marginals of the latent variables.

$$p(\mathbf{x}_t|\mathbf{y}_{1:T}) \quad = \quad p(\mathbf{x}_t|\mathbf{y}_{1:t}, \mathbf{y}_{t+1:T}), \tag{142}$$
$$\propto \quad p(\mathbf{y}_{t+1:T}|\mathbf{x}_t, \mathbf{y}_{1:t})p(\mathbf{x}_t|\mathbf{y}_{1:t}) = p(\mathbf{y}_{t+1:T}|\mathbf{x}_t)p(\mathbf{x}_t|\mathbf{y}_{1:t}) = \beta(\mathbf{x}_t)\alpha(\mathbf{x}_t), \tag{143}$$
$$\propto \quad \mathcal{N}(\mathbf{x}_t|\boldsymbol{\omega}_t, \Upsilon_t) \tag{144}$$

where

$$\Upsilon_t^{-1} \quad = \quad \Psi_t^{-1} + \Sigma_t^{-1}, \tag{145}$$
$$\boldsymbol{\omega}_t \quad = \quad \Upsilon_t(\Psi_t^{-1}\boldsymbol{\eta}_t + \Sigma_t^{-1}\boldsymbol{\mu}_t). \tag{146}$$

We also need to compute pairwise marginals of latent variables, given by

$$p(\mathbf{x}_t, \mathbf{x}_{t+1}|\mathbf{y}_{1:T}) \quad = \quad p(\mathbf{x}_t, \mathbf{x}_{t+1}|\mathbf{y}_{1:t}, \mathbf{y}_{t+1}, \mathbf{y}_{t+2:T}),$$
$$\propto \quad p(\mathbf{y}_{t+1}, \mathbf{y}_{t+2:T}|\mathbf{x}_t, \mathbf{x}_{t+1}, \mathbf{y}_{1:t})p(\mathbf{x}_{t+1}|\mathbf{x}_t, \mathbf{y}_{1:t})p(\mathbf{x}_t|\mathbf{y}_{1:t}),$$
$$\propto \quad p(\mathbf{y}_{t+1}|\mathbf{x}_{t+1})p(\mathbf{y}_{t+2:T}|\mathbf{x}_{t+1})p(\mathbf{x}_{t+1}|\mathbf{x}_t)p(\mathbf{x}_t|\mathbf{y}_{1:t}),$$
$$\propto \quad \beta(\mathbf{x}_{t+1})\exp\left(< \log(p(\mathbf{y}_{t+1}|\mathbf{x}_{t+1})p(\mathbf{x}_{t+1}|\mathbf{x}_t)) >_{q_\theta(\theta)}\right)\alpha(\mathbf{x}_t), \tag{147}$$

which are jointly Gaussian

$$p\begin{pmatrix}\mathbf{x}_t \\ \mathbf{x}_{t+1}\end{pmatrix} \quad = \quad \mathcal{N}\left(\begin{bmatrix}\boldsymbol{\omega}_t \\ \boldsymbol{\omega}_{t+1}\end{bmatrix}, \begin{bmatrix}\Upsilon_t & \Upsilon_{t,t+1} \\ \Upsilon_{t,t+1}^T & \Upsilon_{t+1}\end{bmatrix}\right). \tag{148}$$

To compute the cross-covariance $\Upsilon_{t,t+1}$, we first compute the second derivatives of log of eq. 147 w.r.t. $[\mathbf{x}_t \ \mathbf{x}_{t+1}]^T$:

$$\frac{\partial^2 \log \int d\theta q_\theta(\theta) p(\mathbf{x}_t, \mathbf{x}_{t+1} | \mathbf{y}_{1:T})}{\partial[\mathbf{x}_t \ \mathbf{x}_{t+1}]^2} = -\begin{bmatrix} \Sigma_t^{*-1} & -<A>^\top \\ -<A> & \Psi_{t+1}^{-1} + I + W_{t+1} \end{bmatrix}, \tag{149}$$

where

$$W_{t+1} = \frac{\partial^2}{\partial \mathbf{x}_{t+1}^2} < \log p(\mathbf{y}_{t+1} | \mathbf{x}_{t+1}) >_{q(\theta)}, \tag{150}$$

$$= \sum_{s=1}^{p} \left[ \Sigma_{\mathbf{c}_s} + (\boldsymbol{\mu}_{\mathbf{c}_s} + \Sigma_{\mathbf{c}_s} \mathbf{x}_{t+1})(\boldsymbol{\mu}_{\mathbf{c}_s} + \Sigma_{\mathbf{c}_s} \mathbf{x}_{t+1})^T \right] e^{\mathbf{x}_{t+1}^T \boldsymbol{\mu}_{\mathbf{c}_s} + \frac{1}{2} \mathbf{x}_{t+1}^T \Sigma_{\mathbf{c}_s} \mathbf{x}_{t+1} + \mathbf{d}_s}, \tag{151}$$

evaluated at $\mathbf{x}_{t+1} = \boldsymbol{\omega}_{t+1}$. By negating and inverting the matrix in eq. 149, and using the *Schur* complement, we can obtain $\Upsilon_{t,t+1}$,

$$\Upsilon_{t,t+1} = -(\Sigma_t^{*-1} - <A>^T (\Psi_{t+1}^{-1} + I + W_{t+1})^{-1} <A>)^{-1}(-<A>^T)(\Psi_{t+1}^{-1} + I + W_{t+1})^{-1}. \tag{152}$$

**Computing sufficient statistics of latent variables**

Using $q_{\mathbf{x}}(\mathbf{x}_{0:T}^{(i)})$, we can compute the sufficient statistics of latent variables (that are used in M step).

$$W_{A^{(i)}} = \sum_{t=1}^{T} < \mathbf{x}_{t-1}^{(i)} \mathbf{x}_{t-1}^{(i)\top} > = \sum_{t=1}^{T} \Upsilon_{t-1}^{(i)} + \boldsymbol{\omega}_{t-1}^{(i)} \boldsymbol{\omega}_{t-1}^{(i)\top}, \qquad S_{A^{(i)}} = \sum_{t=1}^{T} < \mathbf{x}_{t-1}^{(i)} \mathbf{x}_t^{(i)\top} > = \sum_{t=1}^{T} \Upsilon_{t-1,t}^{(i)} + \boldsymbol{\omega}_{t-1}^{(i)} \boldsymbol{\omega}_t^{(i)\top}, \tag{153}$$

$$S_{C^{(i)}} = \sum_{t=1}^{T} < \mathbf{x}_t^{(i)} > \mathbf{y}_t^{(i)\top} = \sum_{t=1}^{T} \boldsymbol{\omega}_t^{(i)} \mathbf{y}_t^{(i)\top}. \tag{154}$$

# Hyperaparameter estimation

We take the derivatives of the variational lower bound w.r.t. each hyperparameter to obtain update rules. The lower bound is simplified as below:

$$
\begin{aligned}
\log p(\mathbf{y}_{1:T}^{(1:r)}) &\geq \int d\theta \, d\mathbf{x}_{0:T}^{(1:r)} \, q(\theta, \mathbf{x}_{0:T}^{(1:r)}) \, \log \frac{p(\theta, \mathbf{x}_{0:T}^{(1:r)}, \mathbf{y}_{1:T}^{(1:r)})}{q(\theta, \mathbf{x}_{0:T}^{(1:r)})}, \\
&= \int d\theta \, d\mathbf{x}_{0:T}^{(1:r)} \, q(\theta, \mathbf{x}_{0:T}^{(1:r)}) \, \log p(\mathbf{x}_{0:T}^{(1:r)}, \mathbf{y}_{1:T}^{(1:r)} | \theta) - \int d\mathbf{x}_{0:T}^{(1:r)} \, q(\mathbf{x}_{0:T}^{(1:r)}) \, \log q(\mathbf{x}_{0:T}^{(1:r)}) + \int d\theta \, d\mathbf{x}_{0:T}^{(1:r)} \, q(\theta, \mathbf{x}_{0:T}^{(1:r)}) \, \log \frac{p(\theta)}{q(\theta)}, \\
&= \sum_{i=1}^{r} \log Z'_{(i)} + \int d\theta \, d\mathbf{x}_{0:T}^{(1:r)} \, q(\theta, \mathbf{x}_{0:T}^{(1:r)}) \, \log \frac{p(\theta)}{q(\theta)}, \tag{155}
\end{aligned}
$$

where the line is true from eq. 114, i.e.,

$$-\int d\mathbf{x}_{0:T}^{(1:r)} q_{\mathbf{x}}(\mathbf{x}_{0:T}^{(1:r)}) \log q_{\mathbf{x}}(\mathbf{x}_{0:T}^{(1:r)}) = -\int d\mathbf{x}_{0:T}^{(1:r)} q_{\mathbf{x}}(\mathbf{x}_{0:T}^{(1:r)}) \mathbb{E}_{q_\theta(\theta)} \log p(\mathbf{x}_{0:T}^{(1:r)}, \mathbf{y}_{1:T}^{(1:r)} | \theta) + \sum_{i=1}^{r} \log Z'_{(i)}. \tag{156}$$

So, we need to consider the second term in RHS of eq. 155 for hyperparameter update (the integration w.r.t. $\mathbf{x}$ is omitted, since the integrand is independent of $\mathbf{x}$)

$$\int d\mathbf{a}^{(1:r)} \, dC \, q(\mathbf{a}^{(1:r)}) q(C) \, \log \frac{p(\mathbf{a}^{(1:r)}, C)}{q(\mathbf{a}^{(1:r)}) q(C)} = -KL(C) - KL(\mathbf{a}^{(1:r)}), \tag{157}$$

The first term, $KL(C)$ is given by [1]

$$
\begin{aligned}
KL(C) &= \int dC q_C(C) \log \frac{q_C(C)}{p(C|\gamma)}, \\
&= \sum_{s=1}^{p} \int d\mathbf{c}_s\, \mathcal{N}(\mathbf{c}_s|\boldsymbol{\mu}_{\mathbf{c}_s}, \Sigma_{\mathbf{c}_s}) \log \frac{\mathcal{N}(\mathbf{c}_s|\boldsymbol{\mu}_{\mathbf{c}_s}, \Sigma_{\mathbf{c}_s})}{\mathcal{N}(\mathbf{c}_s|\mathbf{0}, \gamma^{-1}I)}, \\
&= \sum_{s=1}^{p} \left( -\tfrac{1}{2} \log |\gamma \Sigma_{\mathbf{c}_s}| + \tfrac{1}{2} \mathrm{Tr} \left[ \gamma (\Sigma_{\mathbf{c}_s} - \gamma^{-1}I + \boldsymbol{\mu}_{\mathbf{c}_s} \boldsymbol{\mu}_{\mathbf{c}_s}^T) \right] \right).
\end{aligned}
\tag{159}
$$

The first derivative expression w.r.t. $\gamma$ gives us the following update:

$$
\gamma^{-1} = \frac{1}{p} \sum_{s=1}^{p} \mathrm{Tr}[\Sigma_{\mathbf{c}_s} + \boldsymbol{\mu}_{\mathbf{c}_s} \boldsymbol{\mu}_{\mathbf{c}_s}^T],
\tag{160}
$$

Similarly, the second term is given by

$$
\begin{aligned}
KL(\mathbf{a}^{(1:r)}) &= \int d\mathbf{a}^{(1:r)}\, q_{\mathbf{a}}(\mathbf{a}^{(1:r)}) \log \frac{q_{\mathbf{a}}(\mathbf{a}^{(1:r)})}{p(\mathbf{a}^{(1:r)}|\bar{\mathbf{a}}, \sigma^2, \tau^2)}, \tag{161} \\
&= \int d\mathbf{a}^{(1:r)}\, \mathcal{N}(\mathbf{a}^{(1:r)}|\boldsymbol{\mu}_{\mathbf{a}}, \Sigma_{\mathbf{a}}) \log \frac{\mathcal{N}(\mathbf{a}|\boldsymbol{\mu}_{\mathbf{a}}, \Sigma_{\mathbf{a}})}{\mathcal{N}(\mathbf{a}^{(1:r)}|\bar{\mathbf{a}}^{(1:r)}, K)}, \tag{162} \\
&= -\tfrac{1}{2} \log |K^{-1}\Sigma_{\mathbf{a}}| + \tfrac{1}{2} \mathrm{Tr}\left[K^{-1}\Sigma_{\mathbf{a}}\right] + \tfrac{1}{2}(\boldsymbol{\mu}_{\mathbf{a}} - \bar{\mathbf{a}}^{(1:r)})^\top K^{-1}(\boldsymbol{\mu}_{\mathbf{a}} - \bar{\mathbf{a}}^{(1:r)}) + const. \tag{163}
\end{aligned}
$$

The first derivative w.r.t. $\bar{\mathbf{a}}$ is given by

$$
\begin{aligned}
\frac{\partial}{\partial \bar{\mathbf{a}}} KL(\mathbf{a}^{(1:r)}) &= \frac{1}{2} \frac{\partial}{\partial \bar{\mathbf{a}}} (\boldsymbol{\mu}_{\mathbf{a}} - \bar{\mathbf{a}}^{(1:r)})^\top K^{-1}(\boldsymbol{\mu}_{\mathbf{a}} - \bar{\mathbf{a}}^{(1:r)}), \tag{164} \\
&= \frac{1}{2} \frac{\partial}{\partial \bar{\mathbf{a}}} (\boldsymbol{\mu}_{\mathbf{a}} - E\bar{\mathbf{a}})^\top K^{-1}(\boldsymbol{\mu}_{\mathbf{a}} - E\bar{\mathbf{a}}) \tag{165}
\end{aligned}
$$

where $E = \mathbf{1}_r \otimes I_{k^2}$, and this gives us the update rule:

$$
\bar{\mathbf{a}} = (E^\top K^{-1} E)^{-1} (E^\top K^{-1} \boldsymbol{\mu}_{\mathbf{a}}).
\tag{166}
$$

The first derivative w.r.t. kernel parameters (denoted by $\boldsymbol{\alpha} = \{\sigma^2, \tau^2\}$) is given by

$$
\begin{aligned}
\frac{\partial}{\partial \boldsymbol{\alpha}} KL(\mathbf{a}^{(1:r)}) &= \tfrac{1}{2} \mathrm{Tr}\left(K^{-1}\frac{\partial K}{\partial \boldsymbol{\alpha}}\right) - \tfrac{1}{2} \mathrm{Tr}\left(K^{-1}\frac{\partial K}{\partial \boldsymbol{\alpha}} K^{-1}(\Sigma_{\mathbf{a}} + (\boldsymbol{\mu}_{\mathbf{a}} - \bar{\mathbf{a}}^{(1:r)})(\boldsymbol{\mu}_{\mathbf{a}} - \bar{\mathbf{a}}^{(1:r)})^\top)\right), \tag{167} \\
&= \tfrac{1}{2} \mathrm{Tr}\left(K^{-1}\frac{\partial K}{\partial \boldsymbol{\alpha}}(I - K^{-1}(\Sigma_{\mathbf{a}} + (\boldsymbol{\mu}_{\mathbf{a}} - \bar{\mathbf{a}}^{(1:r)})(\boldsymbol{\mu}_{\mathbf{a}} - \bar{\mathbf{a}}^{(1:r)})^\top))\right), \tag{168}
\end{aligned}
$$

where the first derivative of $K(i, j)$ w.r.t. $\boldsymbol{\alpha}$ is given by

$$
\frac{\partial}{\partial \tau^2} K(i, j) = \frac{1}{2\tau^4}(i - j)^2(\sigma^2 + \epsilon\delta_{ij}) \exp\left(-\frac{1}{2\tau^2}(i - j)^2\right) I_{k^2} = \frac{1}{2\tau^4}(i - j)^2 K(i, j),
\tag{169}
$$

$$
\frac{\partial}{\partial \sigma^2} K(i, j) = \exp\left(-\frac{1}{2\tau^2}(i - j)^2\right) I_{k^2}.
\tag{170}
$$

We update $\boldsymbol{\alpha}$ numerically using the derivative expression above.

$$
KL(\tilde{\boldsymbol{\mu}}, \tilde{\Sigma} || \boldsymbol{\mu}, \Sigma) = -\tfrac{1}{2} \log |\tilde{\Sigma}\Sigma^{-1}| + \tfrac{1}{2} \mathrm{Tr}\left[\Sigma^{-1}(\tilde{\Sigma} - \Sigma + (\tilde{\boldsymbol{\mu}} - \boldsymbol{\mu})(\tilde{\boldsymbol{\mu}} - \boldsymbol{\mu})^T)\right].
\tag{158}
$$

# Illustration with simulated data

**A**. Non-stationary population activities

**B**. Correlation coefficients

**C**. Off-diagonal of A

**Figure 2. Illustration of non-stationarity in population dynamics (data simulated from Model II). A**: Raster plots of spontaneous activity from 40 neurons during 10 seconds of recording for simulated trials 1 and 100. We assumed that the two sub-populations (blue and red) have negative correlation at trial 1 and positive correlation at trial 100. **B**: Recovered correlations. Our Model II (red) accurately recovers the correlations between two groups across trials (RMSE: 0.04), while other methods perform poorly: independent PLDSs fit to each trial individually give noisy results (RMSE 0.06) and a single PLDS fit across all trials cannot capture the change in correlation (RMSE 0.44). **C**: Estimation of off-diagonal in dynamics matrices. We fixed the loading matrix $C$ to its true value to avoid issues with non-identifiability of parameters in LDS models. The off-diagonal term $A_{12}$ estimated by our model matched the true values well, whereas the independent PLDS produced noisy estimates, and the fixed PLDS cannot capture the change in $A_{12}$.

We tested Model II using a simulation of spontaneous activity from a population of 40 neurons (simulated from Model II). We assumed that the population could be split into two sub-populations of size 20 neurons each, and simulated an experiment in which the correlation across the two sub-populations changed dramatically across the experiment: Specifically, we generated a 2-d latent state that controls correlations in firing rates between the two groups of neurons, and adjusted the off-diagonal term in the dynamics matrix ($A_{12}$) such that the correlation between the groups varied slowly from $-1$ to $1$ across 100 trials, where the length of each trial is $T = 200$. Other elements of $A$ were adjusted such that the stationary covariance of the system was kept constant.

We fit Model II N-PLDS, a single PLDS, and 100 independent PLDSs to the data. Our model accurately recovered the correlation change in $z$ across trials, while the single PLDS was not able to capture the non-stationarity and the independent PLDSs exhibited noisy correlations (Fig. 2). Finally, our model also accurately recovered the off-diagonal parameter $A_{12}$ (Fig. 2 C). For panel $C$ only, we set the loading matrix $C$ to the ground truth value for each of the models (Model II, fixed PLDS, separate PLDSs). LDS models suffer from non-identifiability of parameters, implying that estimated parameters do not necessarily match the true parameters even for perfect model fits.

# Illustration with real data

Finally, we analyzed a dataset of spontaneous activity recorded from a population of 40 neurons from macaque visual cortex. The details of data collection are described in [1] and the data is available from [2]. Using the spike-sorting information provided in the dataset, we selected the spike-cluster with highest signal-to-noise ratio from each recording channel, and out of those 46 units kept the 40 units with highest firing rates. As the original data consisted of one continuous recording of length 15 minutes, we divided the data into 30 'epochs' of length 30 seconds each, and used every 5th epoch (20% of the data) for testing and the rest (80% of data) for training.

In this data, the mean firing rates are almost constant across time, while the correlations increase at the end of the experiment (Fig. 3 A). After estimating the parameters of our N-PLDS (Model II) from the training data, we computed the predictive distribution on the dynamics matrices $A^*$ for the test data. Using these parameters, we drew samples for spikes to compute the mean firing rates for each trial (Fig. 3 A), as well as the mean pairwise cross-correlations across all neuron pairs. The correlations estimated from N-PLDS (Model II) matched those in the data. For PLDS with fixed parameters, the estimated firing rates and correlations are constant across epochs (Fig. 3 B). To quantify these results, we computed the RMSE in the prediction of mean firing rates and mean correlations on test epochs. The RMSEs on mean firing rate estimation for PLDS are $0.0156, 0.0182, 0.0188$ for $k = 1, 2, 4$, respectively, while RMSE of N-PLDS is $0.0080$ ($k = 4$). The RMSE on mean correlation estimation in PLDSs is $0.0138$ (same for $k = 1, 2, 4$) and $0.0087$ ($k = 4$) in N-PLDS.

**Figure 3. Non-stationary population dynamics (data from [1]).** **A**: Summary statistics of samples from N-PLDS (Model II) with non-stationarity dynamics matrix $A$ for different dimensions of latent dynamics ($k = 1, 2, 4$). The top plot shows the mean firing rate of 40 neurons during 30 epochs, showing that there is only a slight systematic drift in mean firing rate. Each dot represents predicted mean firing rates for the held-out data (6 trials). The bottom plot shows the mean correlation of the spike counts. All three N-PLDS models capture the increase in correlation at the end of the experiment, with the $k = 4$ capturing it most accurately. **B**: Comparison to using a PLDS model with fixed parameters ($k = 1, 2, 4$). Both the mean firing rate and correlation in PLDS are constant across epochs. As a consequence, the best RMSE on mean correlation estimation in PLDS is $0.0138$ ($k = 1$) compared to $0.0087$ ($k = 4$) in N-PLDS.

## Footnotes

[1] The formula of KL divergence between two Gaussians is given by: