[Reviews · NeurIPS 2015]

Submitted by Assigned_Reviewer_1

This paper extends Poisson linear dynamical systems (PLDS) to account for the non-stationarity in neural spike trains.

Their method (NPLDS) uses a hierarchical framework to find the latent variables for each trial, and also scale those latent variables multiplicatively for each trial.

The latent variables are found with a linear dynamical system, and the inter-trial modulators are enforced to be smooth across trials with a Gaussian process.

To fit the model, the authors devised the "Bayesian Laplacian" propagation and used an iterative procedure, which may be of interest to those outside the neuroscience field.

The results are shown to be more predictive than the previous PLDS method, which suggests the added complexity helps performance.

The results are still preliminary in some sense, as it is unclear how this method can be used to further understand population activity, but this method shows promise for other modelers.

Quality:

I found all aspects of the paper to have high quality (in terms of the work).

I trust the results found, although I cannot confirm without testing the algorithm.

Clarity:

All text was well motivated and clear to follow.

Please see below for small typos, etc. Originality:

While the originality of the idea is not new, the method implementation is, and I believe of interest to other modelers. Significance:

This paper is a nice step up from methods from the last NIPS 2014 conference, in which inter-trial variability was not considered.

I believe this paper will be of interest to the modeling community, as the method is an interesting one to fit, and is a natural accept to NIPS.

However, I question if this method will be of interest to the neuroscience community (specifically, how much the method will actually be used).

The authors (and collaborators) will need to demonstrate the method is useful and returns interpretable results, as the method is too complicated to warrant the risk of spending weeks/months analyzing data with it.

One interesting experiment would be to vary the level of anesthesia across trials, and see if this method's latents finds interpretable fluctuations.

There is also no mention that the code will be released to the community; while this is not necessary for publication in NIPS, this would be in the authors' best interests.

Personally, I would not use this method for two reasons: 1) the method has too many levels of complexity with various assumptions, which makes it tough to interpret the latents and 2) I would hesitate to play around with this model (i.e., spending a day throwing my data at it), as there are too many "knobs" to play with.

Thus, I hope this method can make it into a journal paper with scientific results, and I would be much more excited about it!

Notes while reading:

- the intro was beautifully written; the logic was clear and well-founded 083 - instead of p neurons, perhaps you could use n to match other papers (subjective) 085 - let r = N, to follow other papers (subjective) 090 - I believe this equation can be written more concisely 093 - "d controls the mean firing rate of each cell" you need to make it clear this is across all trials (since later you say the mean firing rate changes across trials, and it's confusing 094 - "latent factors" do you mean x_t here, h(i), or both? 100 - give an example of u_t here.

I also question the use of u_t...what if you would like to work with natural images, etc?

If you are using cos, sin, and a binary variable, do you assume all your neurons are "simple"? Figure 1: you need to say the same stimulus is shown for each trial; do you ever reference figure 1 in the text?

also I would suggesting changing "recording" to "trial"...recording is not used in the text 124 - "mean firing rates vary across trials" I tripped here, because I remembered d should be a fixed mean firing rate; see if you can clarify this more by explicitly saying h(i) 205 - "where used the vectorized" typo 214 - "obtaining the the approximate" typo 255 - it would be nice to define z earlier before referencing Figure 2, it took some time to figure out Fig 2 E...can you also think of the x-axis in terms of the number of trials? 298 - "units not used in the original study" --- why were they included here? 299 - say 4 s trials 300 - I find one problem with this model is that by using an LDS, you incur a "stimulus" signal which the experimenter must define.

This limits the ability of the method to be applied to all types of data (e.g., natural images). 323 - you show RMSE here, why not use log-likelihood? is that approximate or tough to compute? Figure 4 - label both axes for every panel

Summary: This paper will be one of the top for neuroscience papers, as it crafts a new method for analysis to deal with inter-trial variability of spike trains.

The method is sufficiently complicated to be of interest for those not related to neuroscience, and the work extends previous methods in a simple yet novel way.

Submitted by Assigned_Reviewer_2

Manuscript titled "Unlocking neural population non-stationarity using a hierarchical dynamics model" describes using an additional time scale over trials to model (slow) non-stationarities. It adds to the sucessful PLDS model, another gain vector matching the latent dimensions that is constant during each trial. Many neuroscientific datasets indeed show such slow drifts, which could very well be captured by such modeling effort.

This is effectively having a set of factors with time bin size equal to the trial length, but this works only when the trial length is fixed. Also, there could be random inter-trial intervals and aborted trials that wouldn't be captured well with such parametrization. For future work, I suggest including the gain on absolute time scale, and not on trials.

In the manuscript, a GP with squared exponential kernel is used as a prior over this longer time scale dynamics. The authors claim that this is better than a random walk assumption, but I do not see why this kernel would be superior to a random walk GP kernel, or an OU kernel as used in the shorter time scale latent processes. A comparison with (1) single extra gain factor per trial, and (2) GP with OU process would be useful.

By the way, even though your prior is stationary, your posterior can be non-stationary. That applies even to the short-time scale latent processes. For this reason, I don't like the title and line #'s 77-81. Also, to ramble more on the title, I don't see why this is "unlocking" anything or, why there's an emphasis on "hierarchy". In my humble opinion, I'd like to see something like "multiple time scales" in the title.

The approximate posterior is given a form that's fully factorized over the trials. This would not allow inference of the covariance of h across trials. Except then in the "Predictive distributions for test data" section, you use the full analytical posterior covariance assuming mu_h from the Bayesian Laplace propagation inference. Isn't this inconsistent (or at least a heuristic)?

The dataset is from Ecker et al. 2014. I know they have put the dataset online with a warning that it is only for validation of their study. Did you get explicit permission from the original authors to use the dataset? Please indicate in the rebuttal.

How was the "most"-nonstationary and -stationary neurons get sorted? Your most stationary neurons are of low firing rate. Was the posterior over h taken into account for this sorting?

Fig 4B suggests you might have needed less than 7 latent dim for h. As future work, a low-rank version of h might provide a more concise parametrization. Can you show how much variance would be explained of Fig 4B if only 2 or 3 dimensions are used?

Fig 4C is not very informative. Why are oscillatory components not explained well by the models? It seems like a key feature of the dataset is not captured by the model family.

Suggestion: The algorithm is somewhat complex for a naive experimentalist to implement. Would you be willing to make your code as available online once accepted?
Summary: Latent GP prior on two time scales to infer slow non-stationary components. I suggest updating title, filling in some missing details, and getting permission from original experimentalists.

Submitted by Assigned_Reviewer_3

Some extra comments - for the real neurophysiological data, it would be very useful to report the recovered time constant of the GP in real terms, i.e. in seconds, rather than # trials. This would apply to Fig 3A-B, 4A-B.

- On line 64, the comment is made that the model separates slow and fast timescale components into the GP and AR components respectively (unlike the standard PLDS). However, I can't seem to find a clear statement of how this separation manifests. Do these two different components simply mop up what they can, given the statistical assumptions embodied by the two processes? Or is the identifiability resolved based on a bimodal distribution of slow modulatory energy in the Fourier domain? There'd be a definite desire amongst practitioners to identify these components with real signals.
Summary: This submission presents a sophisticated (though technical) treatment of non-stationarity in neural recordings. The readability and accessibility could improve in parts, but this is overall a useful model to publish.

Submitted by Assigned_Reviewer_4

Quality: This paper is of good quality. It is technically sound, it builds on tools from LDS and Gaussian processes, and uses simple approximation methods to deal with intractable inference. The results are promising, and the model comparison is convincing, showing that the proposed method is able to capture non-stationary aspects (slow drifts in firing rate) that are evident in neural data but are missed by simpler methods (ie. LDS that cannot deal with non-stationarity). Limitations are also discussed clearly in the Discussion, and possible solutions outlined.

Clarity: the paper is clearly written, easy to follow, and complete derivations are provided. Originality: the method is not particularly novel, as it combines a number of existing methods (LDS, GP, Laplace propagation), but provides a potentially useful advance. Significance: The proposed method could be a useful tool for those interested in characterizing different timescales of variability in neural responses, and identifying different sources of correlated variability.

REF 23: inverted authors names, surnames REF 27: missing Journal information
Summary: The paper combines Linear Dynamical Systems and Gaussian processes to capture non-stationary dynamics in simultaneous recordings from multiple neurons. A useful technical contribution.

Submitted by Assigned_Reviewer_5

The idea of incorporating the h parameter seems interesting. One naive thought is that probably we can just concatenate all the trials to form a huge time series and then fit a PLDS to the whole time series to capture the across trial variation (probably computationally harder).

As for the simulation, besides the outlier issue at latent dim = 8, I am curious why latent dim = 1 gives a good better RMSE as compared to dim = 2,3,4 for NPLDS. (The RMSE has this S-shape, in contrary to the U-shape that is generally observed). Also why the RMSE of PLDS is basically monotone as a function of latent dimension? Seems that LDS kind of model just doesn't fit this set of data that well?
Summary: good paper which extends the PLDS by adding a trial-specific firing rate parameter, which is then given a GP prior to capture the long-term across-trial variation. Simulation result is interesting but not super convincing

Author Feedback
Author rebuttal: We thank the reviewers for their careful reading of our paper and valuable comments on it.

Rev 2

1) Time index: we here used 'trial index' as a measure of absolute time, but the GP-formalism would also allow for using absolute time of the trial, and thus to deal with missing trials or heterogeneous trial lengths.

2) Choice of kernel for GP: We agree that it would be useful to empirically compare the squared exponential kernel on the longer time scale dynamics with other kernels - thanks for the suggestion.

3) Title: Thank you for your points regarding non-stationarity and the title. The 'hierarchy' refers to the fact that we use two processes at the across and within trial level - we could equally have used 'multiple timescales' instead. Instead of 'unlocking', we could also have written 'identifying', and just went for unlocking for purely aesthetic reasons.

4) "The approximate posterior is given a form that's fully factorized over the trials.": Sorry for the confusion. Factorising posteriors are for parameters, that is, the posteriors over each parameter, e.g., q(a), q(b), and q(h) are factorising. But q(h_1, h_2, \cdot, h_m) is not independent with each other. It is normally distributed with the covariance defined by the kernel matrix.

5) Data: Yes, we do have permission from Ecker et al to use their data-set, and this statement would be added to the paper in case of acceptance.

6) 'most stationary neurons': As noted in line 313 and on, we quantified the 'nonstationarity' of each neuron by first smoothing its firing rate across trials using a kernel of size 10, calculating the variance of the resulting smoothed firing rate estimate. This analysis did not use our model, and it is probably indeed biased to low-firing cells-given that we only used this analysis for visualization, we did not attempt to correct for this bias.

7) Can you show how much variance would be explained of Fig 4B if only 2 or 3 dimensions are used? : Although our current results suggest 2 or 3 latent dimension isn't sufficient to explain the data well, looking at how much variance is explained seems a sensible thing to do. Thanks for the suggestion.

8) Oscillatory components in Fig 4C: We agree that there is still room for improvement, but our model still does a lot better than the conventional PLDS in terms of explaining the auto-covariance, which is the point we wanted to make. 

9) Code: We will make code available upon acceptance.

Rev 3

Thanks for the nice evaluation of our work. We will correct the errors that the reviewer noted.

Rev 5

1) concatenate all the trials to form a huge time series and then fit a PLDS to the whole time series : This is indeed the standard approach, and therefore we have included it as "PLDS" in figure 2. For the experimental data (figure 3-4), where we did the fits with left-out trials, concatenating non-continous time series induces unwanted bias, therefore we fitted a PLDS model with shared PLDS parameters across trials - similar to figure 2, and as the reviewer suggests - but with the latents independently drawn from the prior at the start of each trial.

2) The RMSE has this S-shape, in contrary to the U-shape : We would like to emphasise that our intended take-home message of Figure 3 is that there are certain sets of latent dimensions that perform better than others and quantification of those gains, as opposed to finding a single magic number for dimension of h.

Rev 6

1) Report the recovered time in seconds: We will add that in our final version. Thanks for the suggestion.

2) How does the model separate slow and fast timescales: All dynamics that happen within trials are indeed captured by the AR-components, and inter-trial dynamics are captured by the GP term.

Rev 7 

Thank you for your positive assessment of the method.

Rev 8

Thanks for the good feedback and detailed comments on misleading notations/typos in our equations and figures. We will update our manuscript accordingly. We will also release our implementation at the time of publication, and add the link to code in our final version. In our implementation, we will also try to minimise the knob tuning as much as we can, so that more neuroscientists use the package without worrying about initialisation of parameters.

1) Why RMSE not use log-likelihood?: The lower bound under our model with approximations isn't analytically tractable, and comparison of lower-bounds can be inconclusive (as one does not know how tight the bounds are).

2) Do we assume simple neurons: Not necessarily a cell e.g. which has 0 cosine and sine terms would be phase-invariant and hence complex.

3) Why did we include units not used in the original study: The original study used a stringent criterion for excluding neurons which showed any kind of nonstationarity.

4) Nonlinear stimulus dependence: Yes, as the model is written here, it is constrained to linear stimulus dependence.